# The "Ideal Spectrograph" for Atmospheric Observations

Ulrich Platt[1,2], Thomas Wagner[2], Jonas Kuhn[1,2], Thomas Leisner[3]

[1]Institute of Environmental Physics (IUP), Heidelberg University, INF 229,
D-69120 Heidelberg, Germany
[2]Max Planck Institute for Chemistry, Mainz, Germany
[3]Institute for Meteorology and Climate Research, KIT Karlsruhe

**Abstract:**
Spectroscopy of scattered-sunlight in the near UV to near IR spectral ranges has proven to be an extremely useful tool for the analysis of atmospheric trace gas distributions. A central parameter for the achievable sensitivity and spatial resolution of spectroscopic instruments is the étendue (product of aperture angle and entrance area) of the spectrograph, which is at the heart of the instrument. The étendue of an instrument can be enhanced by (1) up-scaling all instrument dimensions or (2) by changing the instrument F-number, (3) by increasing the entrance area, or (4) by operating many instruments (of identical design) in parallel. The étendue can be enhanced by (in principle) arbitrary factors by options (1) and (4), the effect of options (2) and (3) is limited.

We present some new ideas and considerations how instruments for the spectroscopic determination of atmospheric gases could be optimized by using new possibilities in spectrograph design and manufacturing. Particular emphasis is on arrays of massively parallel instruments for observations using scattered-sunlight. Such arrays can reduce size and weight of instruments by orders of magnitude, while preserving spectral resolution and light throughput. We also discuss the optimal size of individual spectrographs in a spectrograph array and give examples of spectrograph systems for use on a (low Earth orbit) satellite including one with sub-km ground pixel size.

## 1 INTRODUCTION

Spectroscopy of scattered sunlight in the near UV to near IR spectral ranges has proven to be an extremely useful tool for the analysis of atmospheric trace gas distributions (see e.g. Platt and Stutz 2008). Applications include the determination of trace gas vertical profiles by MultiAXis Differential Optical Absorption Spectroscopy (MAX-DOAS, see e.g. Hönninger and Platt 2002, Sinreich et al. 2005), observation of volcanic gases, e.g. by the Network for the Observation of Volcanic and Atmospheric Change (NOVAC, see e.g. Galle et al. 2010), and satellite observation of global trace gas distributions (e.g. Burrows, et al. 1996, 1999, Levelt et al. 2006, Veefkind et al. 2012).

Central components of these instruments are moderate resolution (typical spectral resolution, $\lambda/\Delta\lambda$ is around several hundred) grating spectrographs. In all practical applications (except, perhaps observations with direct sunlight) the measurement precision (i.e. the attainable signal/noise ratio, SNR) and thus the detection limit of such spectrographs are ultimately limited by the number of photons detected during a given time interval, i.e. by the spectrographs light throughput (see e.g. Platt and Stutz, 2008). As for most spectroscopic instruments, the light throughput of a spectrograph is basically determined by its étendue.

For example, consider a satellite spectrograph like it is used in the GOME-1/2 (Burrows, et al. 1996, 1999, Munroe et al. 2016), SCIAMACHY (Burrows and Chance 1991, Bovensmann et al. 1999), OMI (Levelt et al. 2006, Dobber et al. 2006), or TROPOMI (Sentinel 5P mission, Veefkind et al. 2012) instruments. These instruments feature ground pixel sizes from 320 x 40 $km^2$ (GOME-1), 80 x 40 km² (GOME-2), 60 x 30 km² (SCIAMACHY), 13 x 24 $km^2$ (OMI) down to 5.5 x 3.5 $km^2$ (TROPOMI), there is a clear evolution towards smaller ground pixel sizes (mainly driven by increase of the online storage capacities and downlink rates) allowing to monitor smaller and smaller structures in the distribution of trace gases in the atmosphere. For instance, the GOME-2 ground pixel more or less covers an entire mega city while the TROPOMI ground pixel size allows identifying structures and hot-spots within a city. It appears clearly desirable to further shrink the ground pixel size, for instance in order to fully resolve industrial or volcanic emission plumes or to identify small scale events. From a high spatial resolution (of e.g. 1 km x 1 km or better) in the spectral range 270-500 nm not only the $NO_2$ retrievals would benefit, but also the retrieval of $SO_2$, BrO, IO, OClO, HCHO, $O_4$, glyoxal, and water vapour. This improvement in spatial resolution can be accomplished for instance by using a longer focal length telescope. When the F-number (ratio of focal length f to diameter of the optics, D) of the telescope is preserved the étendue of the instrument per pixel will not change. Unfortunately, there is a problem: Many more pixels have to be observed. Assuming a 2600 km swath of the instrument (required to obtain global coverage from a sun synchronous low earth orbit within one day) at the satellite velocity of ≈7 km/s an area of about 18200 $km^2$ has to be observed every second. Dividing this area in 5.5 by 3.5 $km^2$ pixels (as TROPOMI does for the near UV and vis bands) requires 743 pixels while 18200 pixels of 1 x 1 $km^2$ would be required i.e. about 24 – times more.

At a given spectrograph size, thus fewer photoelectrons per pixel would be recorded, leading to higher photoelectron shot noise, since the SNR is inversely proportional to the square root of the total number $n_P$ of photoelectrons recorded by a detector pixel (In modern detectors read-out noise and dark current noise are ususally negligible compared to the photon shot noise, see e.g. Platt and Stutz 2008). This geometrical relationship can not be compensated by longer exposure times $\tau_{exp}$, since the orbital velocity $v_{sat}$ of the satellite is fixed. In fact, quite the contrary is true: The along track dimension of the ground pixels are given by $v_{sat} \cdot \tau_{exp}$ (neglecting the along track extension of the instantaneous field of view). Thus smaller along-track extensions of ground pixels require reduced exposure times. A lower SNR of the intensity directly translates into a reduced SNR of the trace gas column density derived from the recorded spectra. Up to now this decrease in SNR at higher spatial resolution was partly compensated by higher trace gas column densities seen by smaller ground pixels. This effect is due to the 'smearing out' of column-density hot-spots by larger ground pixels. However, future instruments with even smaller ground pixel sizes, with spatial extensions comparable to or smaller than the extension of trace gas hot spots, will benefit less or not at all from this effect. Therefore, it is important that future, high spatial resolution instruments will exhibit higher étendue per pixel.

In the following we discuss the design options to maximise the étendue of a spectrograph or spectrograph array. We present new ways to reduce volume and mass of spectrographs for environmental remote sensing applications, while retaining spectral resolution and light throughput. Alternatively, in the same manner, the light throughput can be greatly enhanced without increasing volume and mass of the instrument. Our considerations are largely theoretical and are based on first principles, like the well-known scaling laws of nature as e.g. spelled out by Haldane (1927). However, we neither intent to present a plan for actually realizing an array of spectrographs nor are our considerations restricted to satellite instruments. We also note here that massively parallel optics is used in other areas of science, e.g. in astronomy (see for instance Schilling, 2021).

## 2    SPECTROMETERS FOR DOAS INSTRUMENT - FUNDAMENTALS

Typically DOAS instruments use small to medium-size (focal length f = 50 to 500 mm) grating spectrographs with spectral resolutions in the 0.1 to 1 nm range (see e.g. Platt and Stutz 2008).

### 2.1    Typical design of a DOAS spectrograph

Frequently the Czerny Turner (Czerny and Turner, 1930) design is employed as sketched in Fig. 1A. However, other designs, e.g. imaging grating spectrographs, are also in use (see e.g. General et al. 2014 or Ferlemann et al. 2000, see Fig. 1B). The considerations presented in the following are largely independent of the particular spectrograph design. We also note that modern spectrographs using focal plane detector arrays, which simultaneously record the intensity of the entire spectrum of interest, enjoy the 'multiplex advantage' over a scanning spectrograph. Therefore using interferometers, which may (see Fellgett 1949) or may not (see Barducci et al. 2011) feature a multiplex advantage, instead of grating spectrographs will probably not per se lead to better light throughput.

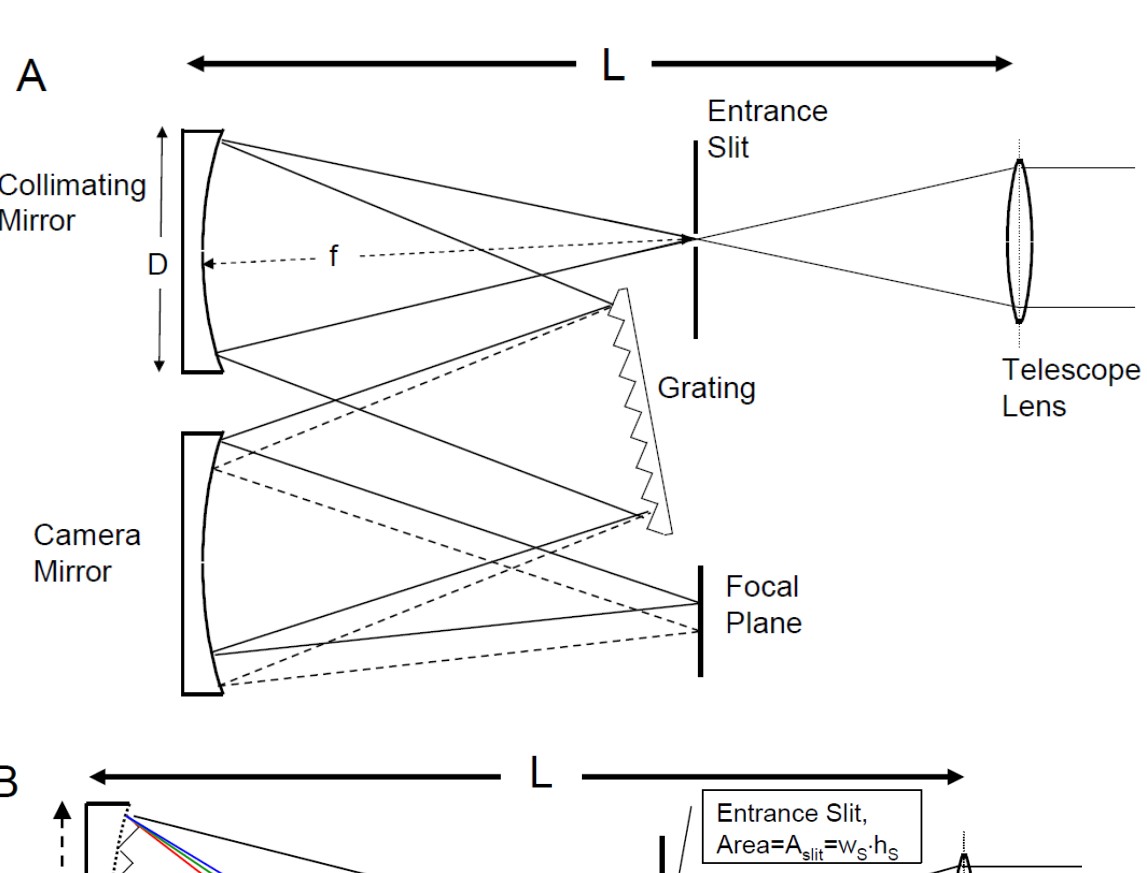

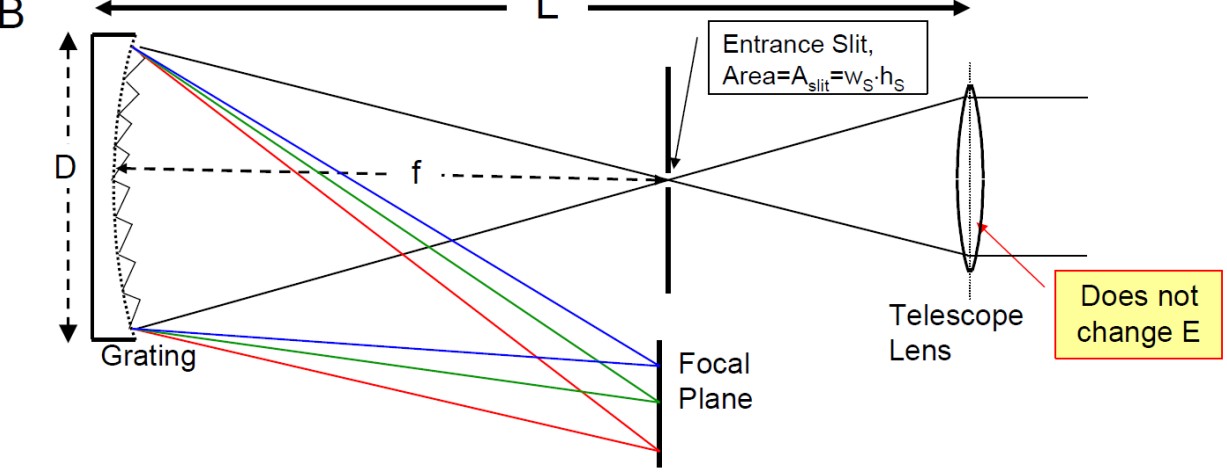

*Fig. 1: Typical design of DOAS Spectrographs with telescope: A) (symmetrical) Czerny Turner spectrograph plus telescope. The size of the spectrograph L is largely determined by the focal length f. Its F-number is given by the ratio of focal length f and diameter of the optics, D. The étendue is a function of the F-number and proportional to the slit area, which is a product of slit width ($w_S$, extension in dispersion direction) and slit height ($h_S$). B) Imaging spectrographs with a concave grating, L and the F-number (see Eq. 6) are given in the same way as for the Czerny Turner spectrograph. Note that in both cases the étendue of the spectrograph is the same as that of the spectrograph+telescope lens.*

## 2.2 The spectrograph light throughput and noise

We assume a spectrograph entrance slit with width $w_S$ and height $h_S$, thus an area $A_S = h_S \cdot w_S$ (see Figure 1), also we assume the aperture solid angle to be $\Omega$. The étendue E of the instrument is thus given by:

$$E = \Omega \cdot A_S = \Omega \cdot w_S \cdot h_S \tag{1}$$

Let's consider a modern compact spectrograph (as e.g. described by General et al., 2014) with an entrance area (i.e. width × height of the spectrograph entrance slit) of $A_{slit} \approx 0.6$ mm$^2$ at an F-number of 4 (see definition of the F-number in Eq. 6, below), equivalent to $\Omega \approx 0.05$ sr. The total étendue (product of free entrance area and solid angle of acceptance of the entrance optics) $E = A_{entr} \cdot \Omega$ of such an instrument would be about 0.03 mm$^2$sr (or $3 \cdot 10^{-8}$ m$^2$sr, see also section 2.3 below).

We further assume the spectrograph to be equipped with a linear detector array, with pixels of width $w_{Pix}$ and a pixel 'height' (pixel dimension perpendicular to the dispersion direction) sufficient to collect all light. The spectral interval $\delta\lambda$ covered by a detector pixel will then be given by the spectrograph's linear dispersion $dx/d\lambda$ and $w_{Pix}$ as: $\delta\lambda = dx/d\lambda \cdot w_{Pix}$. Note that the spectral interval of a pixel is typically by a factor of 2 to 6 smaller than the spectral resolution of the instrument.

Measurements of the clear-sky photon flux F at 320 nm indicate $F \approx 20$ mW m$^{-2}$ sr$^{-1}$ nm$^{-1}$ (e.g. Blumenthaler et al. 1996) at a 30° observation elevation angle and at a solar zenith angle of 68°. The corresponding number of photons registered per pixel and second by such a spectrograph is given by (see e.g. Stutz and Platt 2008 or Platt et al. 2015):

$$\frac{\Delta N_P}{\Delta t} = \frac{E \cdot F \cdot \delta\lambda}{W_{Phot}} \tag{2}$$

Where $W_{Phot}$ denotes the energy of a single photon (about $6.4 \cdot 10^{-19}$ J for $\lambda = 320$ nm). Assuming a typical $\delta\lambda \approx 0.1$ nm and the values for E and F from above results in about $10^8$ photons per pixel and second.

## 2.3 Improving the spectrograph light throughput

In the following, we investigate measures to improve the spectrograph light throughput.

In principle improving the quality of the optics (reflectivity of the mirrors, grating efficiency, etc.) will increase the light throughput, however typically the instruments are rather optimised in this respect and the possible gain due to these measures is rather small (say of the order of 2). Moreover, improved optics can be combined with all measures to be described below. Therefore we restrict our discussion to other measures.

Overall, there are the following options (which to some extent can be combined):

1) Scale the size of the spectrograph, i.e. all three dimensions length $L_1$, height $L_2$, and width $L_3$, and thus the entrance slit are area, while keeping the acceptance angle (i. e. the spectrograph F-number) constant. We refer to this option as 'spectrograph size scaling'

2) Increase the Etendue while keeping some dimensions of the spectrograph constant. For instance scale the acceptance angle (i. e. the F-number) of the spectrograph, while keeping its entrance area $A_S$ constant. We refer to this option as 'spectrograph F-number scaling'.

3) Alternatively the entrance area $A_S$ may be scaled up while retaining the F-number. For instance the slit height $h_S$ could be made larger.

4) Scale number of spectrograph, i.e. use multiple spectrographs with given étendue in parallel and electronically combine the resulting spectra. The latter point will be discussed in more detail below.

Below we have a closer look at the effects of the above options for the improvement of light throughput – while keeping the resolution constant - on spectrograph volume and mass.

### 2.3.1 Spectrograph Size Scaling

We now investigate how the spectrograph light throughput changes when the spectrograph size is scaled up or down while keeping the acceptance angle (i.e. the spectrograph F-number) constant. We assume that the typical dimension of the spectrograph L (e.g. the length of the housing) is scaled from its initial value $L_0$ to some other value $L=\Gamma \cdot L_0$ in such a way that all other dimensions (including entrance slit dimensions) are scaled proportional to L as sketched in Fig. 2, i.e. $L_1$ is scaled to $\Gamma \cdot L_1$ , $L_2$ is scaled to $\Gamma \cdot L_2$, $L_3$ is scaled to $\Gamma \cdot L_3$, $w_{S0}$ is scaled to $\Gamma \cdot w_{S0}$, and $h_{S0}$ is scaled to $\Gamma \cdot h_{S0}$.

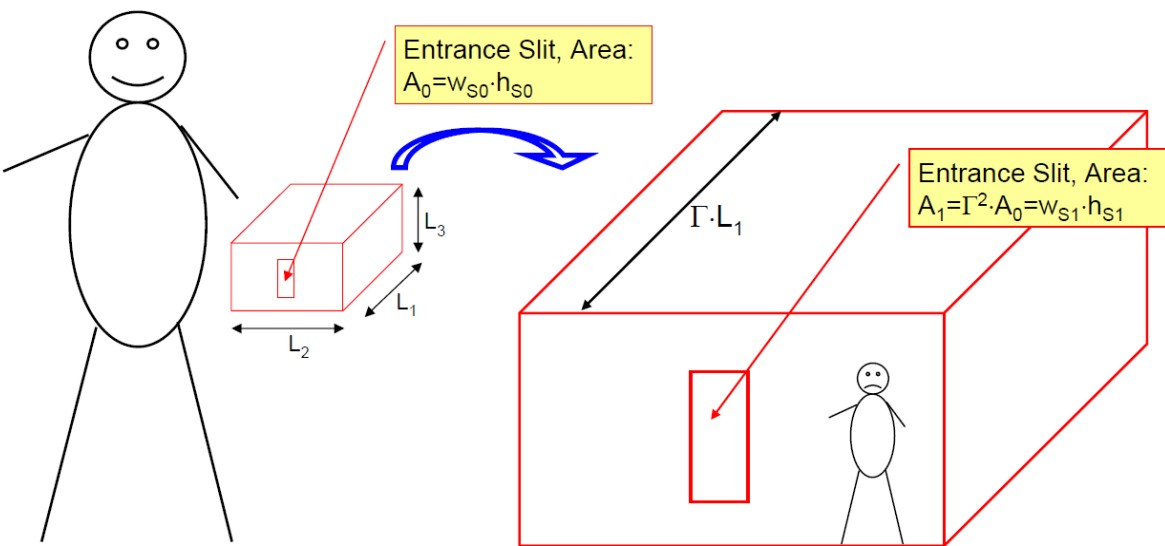

*Fig. 2: Scaling a spectrograph with the linear scaling factor $\Gamma$ at constant aspect ratio, i.e. preserving the ratio between the dimensions $L_1$, $L_2$, $L_3$ as well as between $L_1$ and $w_S$ ($w_S$, slit extension in dispersion direction) and $h_S$ .For instance a spectrograph scaled up by a factor of 10 in its linear dimensions would feature a 100-times higher étendue but have 1000-times the mass.*

Thus, the aperture solid angle $\Omega$ (and F-number) will stay constant, however the étendue $E(L)$ will change from its initial value $E_0 = E(L_0)$, since the area $A_s$ of the entrance slit will scale according to:

$$E = \Omega \cdot A_S \left(\frac{L}{L_0}\right)^2 = E_0 \left(\frac{L}{L_0}\right)^2 = E_0 \cdot \Gamma^2 \propto L^2 \tag{3}$$

5 However, volume and mass of the spectrograph scale with $L_3$, i.e.:

$$M \propto V = M_0 \left(\frac{L}{L_0}\right)^3 \propto L^3 \tag{4}$$

Thus mass and volume of a spectrograph scale with its light throughput (as measured by the Étendue) as:

$$M \propto V \propto L^3 \propto E^{\frac{3}{2}} \quad \text{or} \quad E \propto M^{\frac{2}{3}} \tag{5}$$

10 Note that in the case of satellite instruments the size of the entrance slit will influence the instantaneous ground pixel size, for details see section 4.1.

### 2.3.2 Scale Spectrograph Acceptance Angle

Another option for improving the spectrograph light throughput is increasing its acceptance
15 aperture angle $\Omega$ by changing the aspect ratio of the spectrograph. Here, frequently the F-number (F) of the spectrograph is quoted which is related to the diameter D of the optics and its focal length f. F is defined as (see Fig. 1):

$$F = \frac{f}{D} \tag{6}$$

For moderate F-numbers the following (approximate) relationship between F and $\Omega$ holds:

20 $$\Omega \approx \frac{\pi D^2}{4f^2} = \frac{\pi}{4F^2} \tag{7}$$

Typical DOAS spectrographs have F-numbers between 4 and 6. For satellite instruments in the literature no F-numbers for the actual spectrograph are given, however they can be estimated from the F-number for the telescope (for TROPOMI $\approx 9.5$, see Babic et al. 2019 and Table 2) to be around $F \approx 2$.
25 The corresponding aperture solid angles range from $\Omega \approx 0.2$ (F=2) to $\Omega \approx 0.02$ (F=6).

There are two options (see cases 2a and 2b in Table 1) :
 a) The F-number - for given entrance slit dimensions - could be increased by increasing the area of the mirror (i.e. $D^2$) as given in Eq. 7. This would require to scale D to $\Gamma \cdot D$, $L_2$ to
30   $\Gamma \cdot L_2$ and $L_3$ to $\Gamma \cdot L_3$ as sketched in Fig. 3, while the focal length f and the dimensions of the entrance slit would be unchanged. Since $E = \Omega \cdot A_S$ the volume of the spectrograph would scale as $V \propto \Omega \propto L^2$ (not $L^3$ as in the case of spectrograph size scaling (see subsection 2.3.1 and Eq. 5). Thus, its mass would scale as:

  $$M \propto V \propto L^2 \propto E \quad \text{or} \quad E \propto M \tag{8}$$

35

 b) Alternatively the focal length f could be changed, i.e. from $f_0$ to $f = f_0/\Gamma$. As sketched in Fig. 4. Since changes in f also change the spectral resolution the width of the entrance slit

$w_S$ would have to be changed proportional to f ($w_S \propto f$, i.e. $w_S = w_{S0}/\Gamma$). Therefore the étendue would change as $E \propto 1/f$ (rather than $E \propto 1/f^2$ in the case of constant entrance slit dimensions, as suggested by Eq. 7). In this case the spectrograph mass would scale as:

$$M \propto V \propto \frac{1}{L} \propto \frac{1}{E} \quad \text{or} \quad E \propto \frac{1}{M} \tag{9}$$

Leading to the interesting conclusion that a spectrograph with given spectral resolution but higher étendue would actually be lighter than one with smaller étendue, if the transformation is done by scaling the focal length of the instrument and the width of the entrance slit. While it appears that the two above ways to change the spectrograph aspect ratio are very different and give opposite results, it is easy to show that they are actually the same and can be further broken

down into two steps. This is shown in more detail in appendix 1.

However, the amount of up-scaling the étendue that can actually be applied to a spectrograph in this way is extremely limited due to rapid growth of the imaging errors (e.g. astigmatism) of the optics. Also, the aperture solid angle $\Omega$ of the instrument can usually not exceed (actually not even approach) $2\pi$. Since the entrance area is kept constant (case a) or even shrinks (case b) with

upscaling $\Omega$ the gain in étendue is limited.

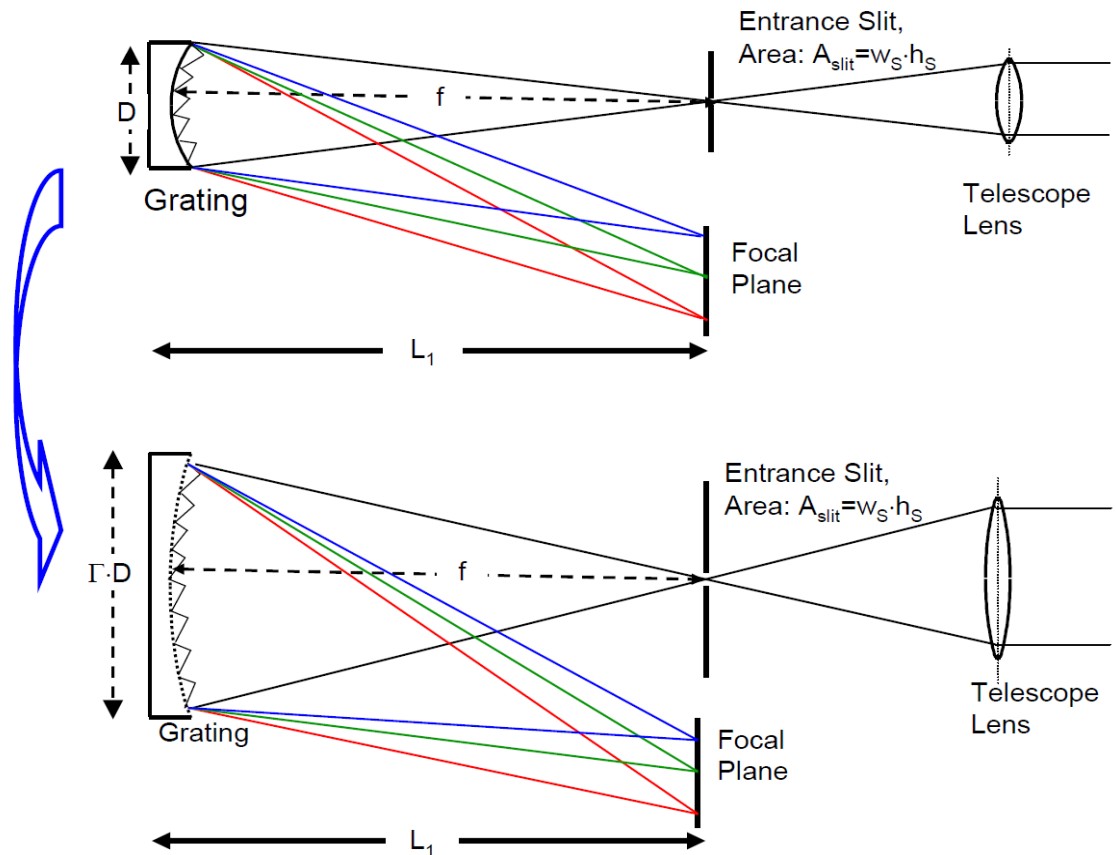

*Fig. 3: Scaling the spectrograph (plus telescope) F-number, option a): Scale size (i.e. diameter)*

*of the optics D with the linear scaling factor Γ, while focal length and slit dimensions, i.e. slit width (extension in dispersion direction) and slit height (extension perpendicular to the dispersion direction) are preserved.*

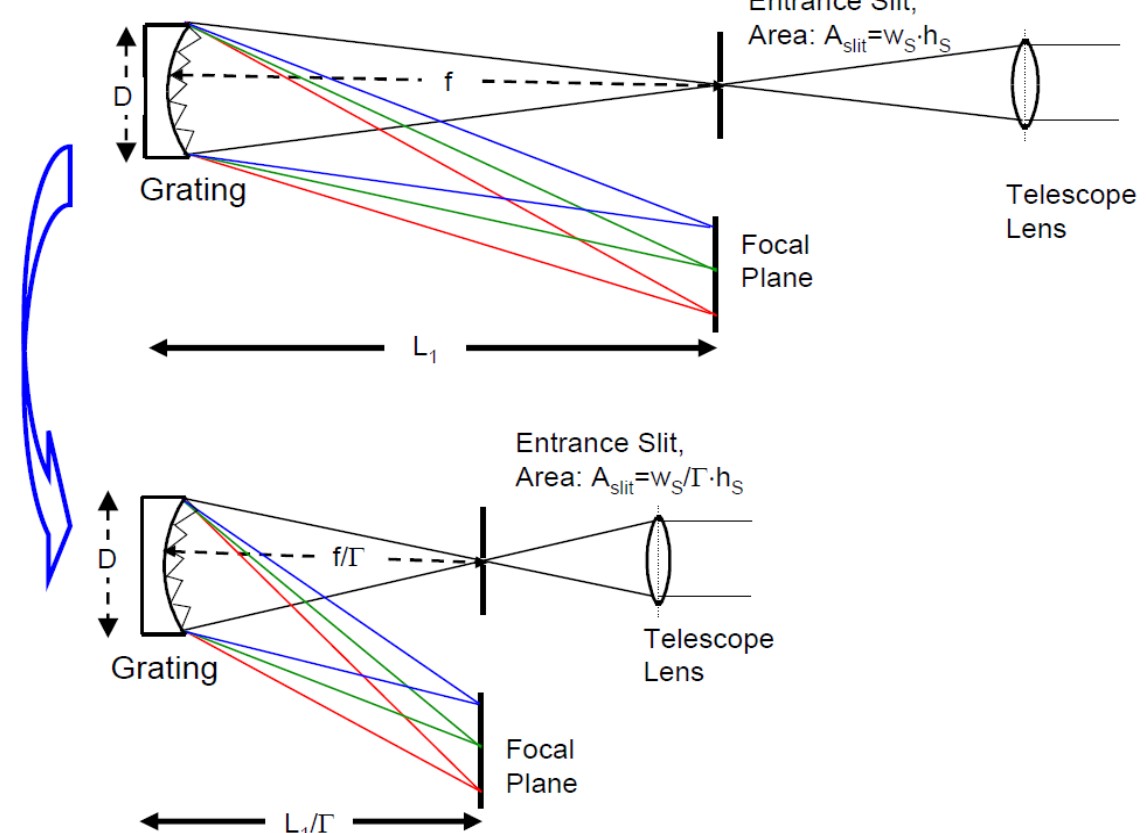

*Fig. 4: Scaling the spectrograph (plus telescope) F-number, option b): Change focal length and entrance slit dimensions (width), while diameter of the optics is preserved. Scaling down the focal length requires narrowing the entrance slit width (reduction of $w_S$) by the same factor in order to preserve the spectral resolution.*

### 2.3.3 Scale Spectrograph Entrance Area

The spectrograph entrance area A is given by $A = w_S \cdot h_S$ (see e.g. Fig. 1). However, widening the entrance slit (i.e. making $w_S$ bigger) at a given spectrograph focal length (and grating grove spacing, see below) would reduce the spectral resolution, so it is not an option. On the other hand the slit height does not seem to have an immediate effect on the resolution, thus increasing $h_S$ (at an otherwise unchanged spectrograph) would appear to be a measure to improve the Etendue. However, there is an increasing amount of image distortion due to astigmatism when $h_S$ is made bigger, which will also degrade the spectral resolution. A quantification of the problem was given by Fastie (1952), who found an empirical relationship between astigmatism as defined as the difference $\Delta f$ between the sagittal focal length and the meridional focal length (see also Kuhn et al. 2021):

$$\Delta f \approx 0.1 \cdot \frac{f}{F^2} \tag{10}$$

The width of the astigmatic spread is then $\Delta L = \Delta f / F$. This corresponds to an additional width of the image $\Delta w$ (in dispersion direction) due to the astigmatism: $\Delta w = \Delta L \cdot h_S / f =$. If

$$\Delta w \approx \Delta L \cdot \frac{h_S}{b} = \frac{\Delta f}{F} \cdot \frac{h_S}{b} = 0.1 \cdot \frac{f}{F^3} \frac{h_S}{b} \underset{b \approx D}{=} 0.1 \cdot \frac{h_S}{F^2} \tag{11}$$

If we allow an additional width $\Delta w = w_S/10$ (and the corresponding slight degradation in spectral resolution) we obtain:

$$\frac{w_S}{10} \approx 0.1 \cdot \frac{h_S}{F^2} \quad \text{or} \quad h_S \approx w_S \cdot F^2 \tag{12}$$

From this consideration it becomes clear that the slit height is limited, for instance for a typical $F = 4$ spectrograph with $w_S = 50 \mu m$ one obtains $h_S \approx 16 \cdot w_S \approx 0.8$ mm. Moreover, smaller F-numbers would require less slit height in order to retain the desired resolution. This relationship severely limits the gain in étendue possible by either reducing the F-number or increasing the slit height.

It should be noted that reducing the grating groove spacing g (all other spectrograph parameters being kept unchanged) can also be a way to improve the étendue of a spectrograph, at least if the spectral range covered by the instrument is not a high priority. A smaller groove spacing g will enhance the linear dispersion of the instrument approximately proportional to 1/g, thus the width of the entrance slit $w_S$ (and its height $h_S$, see Equ. 12 above) can be made proportionally wider, which should enhance the étendue approximately as $E \propto g^{-2}$. This measure is clearly limited, since the grating groove spacing should not be smaller than the wavelength and usually gratings are selected to have a groove spacing close to this limit.

There is one little discussed possibility to further enhance the groove density, which relies of 'immersing' the grating in a transparent (for the wavelength range to be measured) material with an index of refraction n > 1 (see e.g. Larsson and Neuhaus H. 1968).

Thus the grating will see not the vacuum (or air) wavelength $\lambda_0$ but rather $\lambda_0/n$, which can be considerably shorter, allowing proportionally higher groove densities. Possible materials for the UV (and visible) range could be Quartz (n≈1.5-1.6), Sapphire (n≈1.6-1.8), or Diamond (n≈2.4-2.6). In the short-wave infra-red range crystalline silicon (n≈3.5) was successfully used (van Amerongen et al. 2010).

### 2.3.4   Scale number of spectrographs

In a number of applications (e.g. for the satellite instruments GOME, SCIAMACHY, GOME-2, OMI, and TROPOMI, see Introduction) the total spectral range is divided among several spectrographs, each covering part of the total wavelength interval of the instrument. However in all DOAS applications for each spectral interval only a single spectrograph is used.

Up to now the possibility to use a number of $N_{Sp}$ spectrographs (for simplicity assumed to be identical in design, each with Étendue $E_0$) in parallel and co-adding their spectra was not used, although this option clearly enhances the light throughput of the system:

$$E_{tot} = N_{Sp} \cdot E_0 \tag{13}$$

In this case (see case 3 in Table 1) the total mass of such an array of spectrographs (assumed to be of identical design) scales with $N_{Sp}$.

$$M \propto N_{Sp} \quad \text{and} \quad M \propto E \tag{14}$$

Note that this is a more favourable scaling of E with M than in the case of scaling the size of a spectrograph (see Equation 5). For instance, in order to enhance E from $E_0$ to $10 \cdot E_0$ an array of 10 spectrographs would be 10-times more heavy, while scaling up a single spectrograph would end up in an about 32-times heavier instrument.

### 2.3.5 Summary, scaling spectrographs

Table 1 summarizes the above discussed scaling options for improvement of spectrograph light throughput at a given spectral resolution. Changing the focal length (option 2b) appears to be the by far best option since the spectrograph mass is actually reduced when the étendue is improved by reducing the focal length (even when the entrance slit width has to be reduced to maintain the spectral resolution). However, the amount of scaling that can be applied to a spectrograph in this way is extremely limited due to limitations in the imaging optics. The same is true for scaling the mirror area (option 2a), where the mass scales in proportion to the improvement in étendue. Thus scaling the number of spectrographs remains as the most favourable option with the mass scaling in proportion to the improvement in étendue.

*Table 1: Summary of scaling options for the improvement of spectrograph light throughput at a given spectral resolution.*

| Scaled Property | Mass – Étendue relationship | Aspect Ratio Preserved | Comment |
|---|---|---|---|
| 1  Spectrograph size | $M \propto E^{\frac{3}{2}}$  or  $E \propto M^{\frac{2}{3}}$ | Yes | No limit to upscaling |
| 2a  Mirror size (area), F-number | $M \propto L^2 \propto E$  or  $E \propto M$ | No | Very limited scaling, conflict with 3 |
| 2b  Focal length F-number | $M \propto \dfrac{1}{E}$  or  $E \propto \dfrac{1}{M}$ | No | Very limited scaling, conflict with 3 |
| 3  Slit height | $E$ independent of $M$ | Yes | Very limited scaling, conflict with 2 |
| 4  Number of spectrographs | $M \propto N_{Sp}$ and $M \propto E$ | Yes | No limit to scaling |

## 3   SPECTROGRAPH ARRAYS

In the previous section we concluded that scaling the number of spectrographs, i.e. using an array of several spectrographs instead of a single one, is the optimal way to improve the étendue and thus the light throughput of a spectrograph system by a large factor. We also note here that massively parallel optics are used in other areas of science, e.g. in astronomy (see for instance Schilling, 2021). In the following we investigate a number of practical questions associated with the introduction of spectrograph arrays.

### 3.1   Improve the throughput/weight ratio of a spectrograph

If we wish to keep the light throughput constant when scaling (down) the size (given by L) of the instrument we can just use a large number of (ideally) instruments with identical properties in parallel. The spectra of all instruments are then co-added as to keep the light throughput constant. Since $E \propto L^2$ we need to increase the number of individual spectrographs if $L < L_0$. The number $N_{Sp}$ of spectrographs required (which of course needs to be rounded to the nearest integer) will be:

$$N_{Sp} = \left(\frac{L_0}{L}\right)^2 \tag{15}$$

The total mass of an array of spectrographs scaled to $L < L_0$ is then given by:

$$M \cdot N_{Sp} = M_0 \left( \frac{L}{L_0} \right)^3 \cdot \left( \frac{L_0}{L} \right)^2 = M_0 \left( \frac{L}{L_0} \right) \propto L \qquad (16)$$

This means that the mass (and volume) shrink with the scaling if e.g. a single spectrograph with characteristic dimension $L_0$ is replaced by an array of N smaller spectrographs, each one scaled down in its linear dimensions to $L_0/N$.

Thus, it appears that it would be of advantage to use a large number of very small spectrographs in order to reduce volume and weight of an instrument. However, there are limits how far we can shrink a spectrograph, at least as long as we consider conventional spectrograph design.

### 3.2 Is it true that the spectrograph mass scales with $L^3$?

In the above section we assumed that the spectrograph mass scales with the cube of the outer dimension, i.e. a characteristic dimension L. But how will the rigidness of such an instrument change if all dimensions are scaled by the same factor $L/L_0$.

If, for simplicity, we assume the spectrograph to behave like a bar with length L, width w, and height h (see sketch in Fig. 5) on which an external force acts. Then we can apply the famous

case of bending a bar, which is a described in most physics textbooks (see e.g. Meschede 2015). When scaling the initial length $L_0$ of the bar to some other length L by a factor $L/L_0$ and likewise $w_0$ to $w = w_0 \cdot L/L_0$ and $h_0$ to $h = h_0 \cdot L/L_0$ we can calculate the scaling of $\Delta h$ since:

$$\Delta h \propto L^3, \ \Delta h \propto h^{-2}, \ \Delta h \propto w^{-1} \qquad (17)$$

Since w and h are scaled proportional to L we have:

$$\Delta h \propto \frac{L^3}{L^2 \cdot L} = \text{const.} \qquad (18)$$

Thus the bar, respective spectrograph casing will bend by the same absolute amount when subjected to a certain force. We can assume that the bending force actually scales with L as well, because for instance thermal stress as well as external stress, e.g. due to bending of the mounting base plate, is proportional to the dimension L, If we assume Hooke's law to hold we can

conclude that the deformation $\Delta h$ of a spectrograph frame probably scales with 1/L. This, again, means that the performance of a scaled spectrograph will not change with scaling since the requirements for alignment of the optical elements also scale with L. For instance in a scaled down spectrograph the pixel size of the detector array will also shrink.

In conclusion we can say: In first approximation scaling of a spectrograph by changing all

30 dimensions will not change its performance as far as it is determined by the geometry of the instrument and therefore its mass will scale with $L^3$ as assumed above.

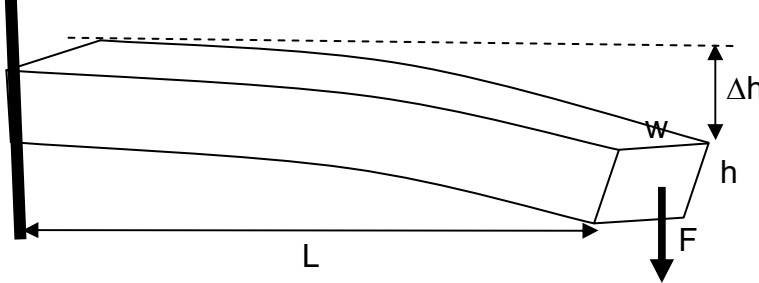

*Fig. 5:Bending of a bar. When a given force F is applied as shown the amount of bending $\Delta h$ is proportional to $L^3$, $1/h^2$, and $1/w$. Scaling all three quantities by the same factor leads to $\Delta h$*

*(at a given force) being independent of L.*

Another point may be the thermal stability, which is a function of the thermal time constant of an object. The thermal time constant is given by the ratio of heat flow, being proportional to the surface of an object, and its heat capacity, which is (for a given material) proportional to the volume of the object. Thus the thermal time constant of a spectrometer should scale with 1/L.
Moreover, it can be assumed that a smaller instrument requires shorter and thus lighter wiring (although it amounts to only a small fraction of the total mass anyway).

### 3.3 How far can we shrink a spectrograph?

Obviously, we can not shrink spectrographs indefinitely since then they will not function any more. In additions to possible mechanical constraints, the following phenomena (see also e.g. Avrutsky et al. 2006) limit the shrinking of spectrographs:

1) Light diffraction at the ever shrinking entrance slit

2) The grating will loose its resolving power.

3) Very small detector pixels are required

1) For a very long rectangular aperture (i.e. the entrance slit) with width $w_S$ (i.e. a slit with $h_S \gg w_S$) the diffracted intensity is given by:

$$I(\vartheta) = I_0 \frac{\sin^2 x}{x^2} \quad \text{with} \quad x = \frac{\pi w_S}{\lambda} \cdot \sin \vartheta \tag{19}$$

Thus the first minimum is at $x = \pi$ with $\sin\vartheta_1 = \lambda/w_S$. In order to use the slit image at least the first diffraction order must hit the collimating mirror (or imaging grating) thus $\sin(\vartheta_1) \approx D/2f = 1/2F = \lambda/w_S$ or $w_S \approx 2 \cdot \lambda \cdot F$. A more precise calculation actually yields $w_S$ being closer to (actually slightly smaller than) $\lambda \cdot F$, thus for $\lambda = 320$ nm and $F = 4$ one obtains $w_S \approx$ around 2 μm.

2) The resolving power $P = \lambda/\Delta\lambda$ of a grating with grating constant G (in grooves/mm) and width $w_G$ (in mm) is given by its total number $N_G$ of grooves:

$$P = \frac{\lambda}{\Delta\lambda} = N_G = \frac{G}{w_G} \tag{20}$$

The smallest spectrographs typically used in (scattered sunlight) DOAS instrument are 'miniature spectrographs' like the Ocean Optics (Ocean Optics 2020) USB2000 or Avantes AvaSpec-Mini (Avantes 2020) instruments featuring $f \approx 70$ mm equipped with an entrance slit with $w_S = 0.050$ mm and $h_S = 0.5$ mm. The F-number of the instruments is about 4, corresponding to an aperture solid angle $\Omega \approx 0.25^2/4 \cdot \pi \approx 0.0491$.
The corresponding étendue will be $1.23 \cdot 10^{-9}$ m$^2$sr (0.00123 mm$^2$sr). The grating typically has 1800 grooves/mm resulting in a total number of 36000 grooves and a theoretical resolving power $P = 36000$. In practice, because of the relatively wide entrance slit, the spectral resolution is about 0.5 nm at 300 nm corresponding to a resolving power $P_{pract} \approx 600$.

3) The detector arrays typically have a pixel pitch around 12 μm. If a spectrograph is to be scaled down also the pixel pitch must be scaled (with the same linear scaling factor). Presently detectors with pixel pitches around 1 μm are mass produced and are used in many consumer products (smartphones, webcams, etc.). Although these sensors are primarily designed for visible light detection it has recently been shown that UV sensing is also possible with these cameras (Wilkes et al. 2017a, b).

In summary: Even rather small 'miniature' spectrographs (like Ocean Optics USB-2000 or Avaspec mini) with focal lengths around $f \approx 50\text{-}70mm$ probably could be scaled down by $L/L_0 \approx 0.1$. Thus, an array of 100 of such micro-spectrographs (+ telescope) could replace a conventional miniature spectrograph at about one tenth of volume and weight. Of course for larger spectrographs as are e.g. used in satellite instruments or active LP-DOAS even higher scaling factors are in principle possible.

## 3.4 Spectrograph Stray Light

Here we have a quick look on the effect of spectrograph-system optimisation on the stray light level. Stray light can have negative effects on the precision of spectroscopic trace gas measurements, as e.g. pointed out by Platt and Stutz (2008). Note that stray light can be comparatively high in spectrographs filtering a relatively broad wavelength interval from a continuous spectrum as in typical DOAS applications. As also pointed out by Platt and Stutz (2008) a typical stray light level of $I_{SL}/I \approx 10^{-5}$ as derived by illuminating the instrument with a monochromatic source (see e.g. Pierson and Goldstein 1989) translates into stray light levels being closer to $10^{-2}$ than to $10^{-5}$.

Sources of stray light include light scattered by the optical elements (grating, mirrors, and the detector surface) of the instrument, reflection of unused diffraction orders off the spectrograph walls, reflection of unused portions of the spectrum from walls near the focal plane, reflections from the detector surface (see e.g Pierson and Goldstein 1989). A further, potentially important source of stray light is due to incorrect illumination of the spectrograph: If the F-number of the illumination exceeds that of the spectrograph radiation will overfill the collimating mirror and hit interior walls of the instrument, from where it may be reflected to the detector.

In general the amount of straylight is proportional to the ratio of the area of the scattering surfaces and the detector area, i.e. basically its amount scales with the inverse of the F-Number. Overall, it appears that the relative amount of stray light should not change when the spectrograph is scaled, such that its aspect ratio (and thus its F-number) remains unchanged (i.e. according to case 1 in Table 1). Of course running an array of spectrographs of identical design in parallel (case 4 in Table 1) should also not affect the relative amount of stray light.

## 3.5 Further considerations

It can be desirable to have a small or vanishing polarisation sensitivity of spectrometers used for the analysis of sunlight reflected from Earth's surface or scattered in the atmosphere. In some satellite instruments, e.g. OMI, TROPOMI, 'polarisation scramblers' are used to reduce the polarisation sensitivity of the spectrometer. There are many different designs of polarisation scramblers (e.g. Lyot depolarizer or wedge depolarizer, which are based on plates consisting of birefringent material being placed in the optical path of the instrument, see e.g. Caron et al. 2012). These devices have in common that they are rather small plates consisting of two wedges (wedge angle around 1°) made of birefringent material. These are placed in the optical path of the instrument, typically at a suitable position between telescope entrance and entrance slit. In the case of TROPOMI two pairs of wedges (quartz and magnesium fluoride, respectively) are used, there volume is around 1 cm$^3$ (Babic et al. 2019).

Obviously, for very small spectrometers and telescopes the depolarizer will also be very small (a fraction of 1 cm$^3$), thus adding negligibly (<3%) to the volume and weight of the instrument.

Another point, which is particularly relevant for satellite applications is the use of direct solar reference spectra. The usual means of obtaining these spectra relies on directing sunlight via diffuser plates into the instrument. This approach, however, faces some technical difficulties due to the spectral structures introduced by the diffuser plates (see e.g. Richter and Wagner , 2001).

Therefore we recommend to apply the ‚reference sector method' or related techniques, which are frequently used and do not rely on diffuser plates.

We also note that smaller instruments require shorter power (or data) connection wires, also thermal ducts can be shorter and thus lighter (see also section 3.2). Similarly, radiation shielding (of e.g. detectors in satellite instruments from cosmic radiation) becomes much lighter for smaller instruments. Alternatively, one might want to keep the mass, and thus the power lines and shielding constant and benefit from an enhanced light throughput. Moreover, using a large number of spectrometers in parallel has the potential advantage that the data from individual detectors, which were affected by cosmic radiation, can be sorted out and are not co-added during the evaluation procedure.

Finally, a spectrograph array design also reduces complexity since a much simpler optical design can be used for each pixel (or small number of pixels), which is just repeated many times (see e.g. section 4.1). In fact in most cases a single (or a few) rather complex instrument(s) are replaced by a large number of comparatively simple instruments of identical design. There will likely be additional effort in cross calibration, which, however may be offset by a simplified design.

### 3.6    How to combine the signal of a large number of spectrographs?

In principle this is a straightforward task: If all individual spectrographs of an array (i.e. set of spectrographs with identical spectral ranges and viewing directions) were truly identical in spectral resolution and spectral registration (wavelength calibration and dispersion) then the detector output signal of corresponding pixels only had to be individually digitized and co-added. How well this prerequisite for simple co-adding is actually met depends on the manufacturing process and its precision for the individual (miniature) spectrographs. If the deviations of the individual spectrographs only amount to a fraction of a pixel one might chose to still simply co-add the spectra and accept a certain degradation in spectral resolution.

If it should be found that the individual spectrographs have considerable individual deviations in spectral registration a correction by shifting and stretching/compressing the individual spectra prior to co-adding might be necessary. These tasks require some effort in post-processing, however with the rapid advancement of electronics and information technology in recent decades this should not be a major problem. For instance advanced bus-systems could be used to interconnect the individual spectrographs.

For satellite instruments (see section 4.1, below) the idea is to basically have one (or a small number of) spectrometer(s) per viewing direction. In the case of a single spectrometer (with 1-dimensional detector) per viewing direction there would be no change in the amount of data generated compared to an approach using a single spectrograph with 2-dimensional detector (as e.g. in the OMI or TROPOMI instruments). In designs where several spectrometers observe the same ground pixel (see section 4.1) data could be co-added on board. Thus, again there would be no increase in data rate compared to a single spectrograph with 2-D detector approach.

### 3.7    How to manufacture arrays of (micro) spectrographs?

Clearly, the wide spread use of arrays of large numbers of (micro-)spectrographs hinges on efficient manufacturing techniques for these instruments. Miniaturized spectrographs based on conventional spectrograph design are described by a number of authors, e.g. Avrutsky et al. 2006, Wilkes et al. 2017, Danz et al. 2019. These authors also mention modern manufacturing techniques.

In particular at present technologies for mass production are available, like 3D printing or automated machining of the frame. Also, the optical alignment of spectrographs can be automated, here replica optics could help. In addition, the required electronics and detectors have become very affordable during recent decades.

Furthermore, in the case of satellite instruments space qualification and documentation of a large number of identical (and rather simple) spectrographs may mean less effort than that of a single or a few (relatively complicated) spectrographs.

## 3.8 Unconventional spectrograph designs

A number of ideas for unconventional spectrograph designs were reported. For instance how to enhance the light throughput of imaging spectrographs (for the visible and near IR spectral range) are presented by Chrisp et al. (2020). Park and Choi (2013) suggested Fresnel optics to miniaturize spectrographs. Furthermore, a completely new principle for spectrograph design was proposed, e.g. by Grundmann (2019a, b), it relies on using a special type of diode array as only element of the instrument. The pixels of the diode array are manufactured in such a way that the bandgap of the semiconductor increases with the pixel number (this is achived by using a binary or ternary semiconductor with a composition varying with the pixel position). The light enters along the long axis of the detector array (which acts as a waveguide) at pixel 1, which has the smallest bandgap and therefore absorbs the longest wavelength radiation while transmitting radiation with shorter wavelength. Pixel 2 has a slightly wider bandgap absorbing radiation with slightly (by $\Delta\lambda$) shorter wavelength and so forth. The resolution of the device is approximately equivalent to $\Delta\lambda$. In a practical device a spectral resolution of 0.01eV at 3.5eV (corresponding to about $\Delta\lambda\approx$1nm at $\approx$355nm) was reached.

## 4 PROPOSAL FOR OPTIMIZED SPECTROGRAPHS

Judging from the above considerations in most applications replacement of existing spectrographs by an array of scaled-down micro spectrographs of identical design (see Fig. 6) would result in considerable (up to two to three orders of magnitude) reduction in volume and weight. As mentioned above, even if miniature spectrographs are taken as basis for comparison an order of magnitude reduction appears possible.

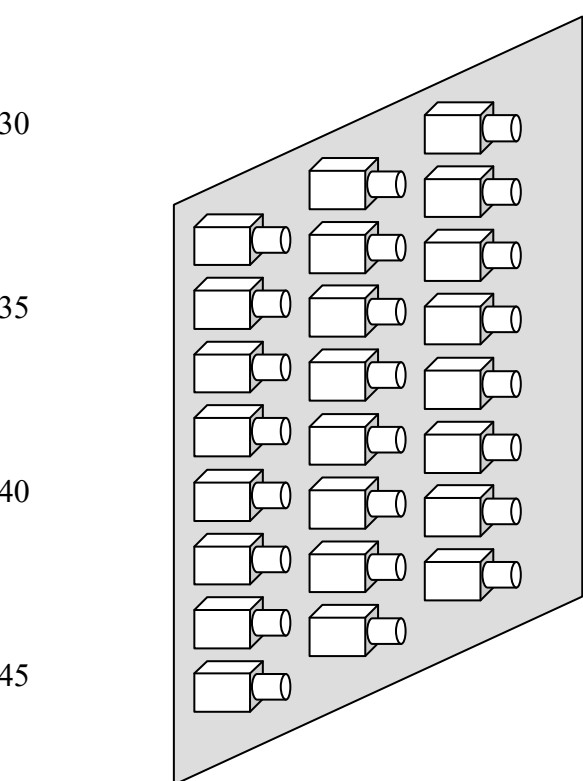

*Fig. 6: Array of spectrograph – telescope combinations of identical design.*

### 4.1 Satellite Applications

There are a number of satellite instruments in orbit (e.g. GOME, GOME-2A, B, C, SCIAMACHY, ODIN, OMI, TROPOMI, OMPS NM, GEMS (geostationary), EMI), which are based on small UV-visible-Near IR spectrographs (typical focal length around 200 mm) coupled to small telescopes (typical diameter 1 cm). The typical telescope field of view angle (for one ground pixel) is around 0.25 to 1 degree.

Usually per wavelength range one spectrograph is used. The total number of spectrographs ranges from 2 (OMI, Levelt et al. 2006), 4 (TROPOMI, Veefkind et al. 2012, Dobber et al. 2006), 4 (GOME and GOME-2, Burrows 1999) to 8 (SCIAMACHY, Burrows and Chance 1991, Goede et al. 1991, Burrows et al. 1995, Bovensmann et al. 1999).

The scanning (i.e. cross-track spatial resolution) is either achieved by a mechanical scanner or by imaging spectrographs (2D-spectrographs where one dimension is devoted to wavelength, the second to space), see e.g. Levelt et al. (2006) or Veefkind et al. (2012). In particular, these imaging instruments are very sophisticated designs featuring extreme properties like very large cross track fields of view combined with extremely small along track aperture angles. These truly remarkable features come at a price: in some cases a-spherical (or even free-form) optics have to be used and only rather large F-numbers are possible.

In order to reduce weight and volume of instruments of this type the single spectrograph (per wavelength range) could be replaced by an array of scaled down spectrographs, each observing one or a few ground pixels. Each spectrograph would have its own telescope, thus cross track resolution could be achieved by aligning the field of view of the individual spectrographs accordingly, as sketched in Fig. 7a. Thereby the advantages of avoiding scanners by using the pushbroom principle are combined with a rather simple design (of the individual telescopes).

In fact, there could be one or several spectrographs per viewing direction and wavelength interval. This approach would have no more drawbacks, for instance with respect to 'destripig' measures, than existing whisk-broom designs (like OMI or TROPOMI). The cause(s) for the 'striping phenomenon' (slight changes in the derived SCD values across the swath) are not fully understood, but they are probably due to somewhat different instrument functions for each viewing direction. On the other hand, such a spectrograph per viewing direction (SPVD) approach could have great advantages besides the obvious possibility of achieving better light throughput and thus SNR:

1) Much simpler spectrometer design, here a conventional Czerny-Turner design or imaging grating design is assumed. Additional light throughput could be gained by the measures described in section 2.3. Obviously the telescope has to be designed in such a way that the projection of the ground pixel matches the spectrograph entrance slit size (see Fig. 7b).

2) Much simpler telescope design, since only a small telescope field of view is required.

3) Adaptive field of view for the edges of the swath (for a daily coverage by a LEO instrument a ≈2600 km swath is needed) in order to reduce the variation in ground pixel size across the swath. At 800 km satellite altitude the pixels at the edge of the swath are roughly twice as long (along track extension) and four times as wide (cross track extension) than in the centre of the swath, i.e. in satellite-nadir direction (see Fig. 7a).

4) More redundancy in the design, the failure of an individual spectrograph would not be catastrophic.

Regarding the ground pixel size, there are three reasons why the ground pixels are larger towards the edges of the swath (assumed here to be 2600 km at a satellite altitude of 800 km):

1) Because the pixels at the edge of the swath are further away from the instrument. This effect enlarges the pixels by a factor of 1.91 in cross-track as well as in along-track direction (area is thus enlarged by a factor of 3.64).

2) The pixels at the edge of the swath are seen under a larger angle ($\approx 58.4^o$) thus their cross track extension (but not the along track extension) is further enlarged by another factor of 1.91 enhancing its cross-track extension to 3.64 over the nadir case (area increases by a factor of 6.94).

3) The larger angle at which the pixels at the edge of the swath are seen is further enlarged due to the curvature of Earth (enlarging the viewing angle at the edge of the swath by $11.5^o$) bringing the total angle to $69.9^o$. Thus the cross track extension of the ground pixel is extended by a factor of 5.56 instead of 3.64 for the flat Earth case (or an additional factor of 1.52). Ultimately the ground pixel area is by a factor of 10.6 larger than in nadir.

This latter effect (curvature of Earth) has the smallest influence on the ground pixel size at the edge of the swath. Thus a ‚flat Earth' approximation could be considered for the sake of simplicity.

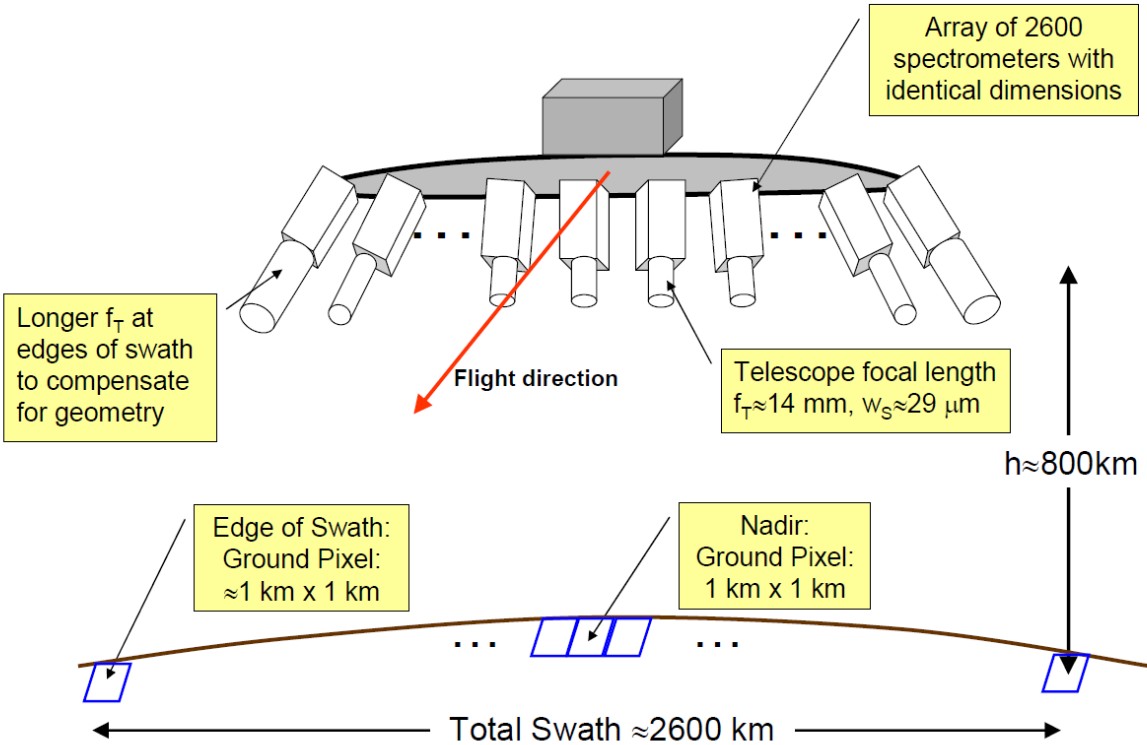

*Fig. 7a: Possible arrangement of an array of spectrograph + telescope combinations for satellite application (this example depicts the 'scaled 2' arrangement, see Table 2). In principle, there is one spectrograph + telescope for a viewing direction (i.e. ground pixel, see Table 2). (In the case of the 'scaled 2' arrangement, two spectrographs observe the same ground pixel to improve the S/N ratio) Longer focal lengths $f_T$ of the telescope observing ground pixels near the edge of the swath could be chosen to compensate for their larger size. For simplicity a linear arrangement of the spectrographs along a line is shown, in practice an arrangement in a 2-D array would of course be much more compact.*

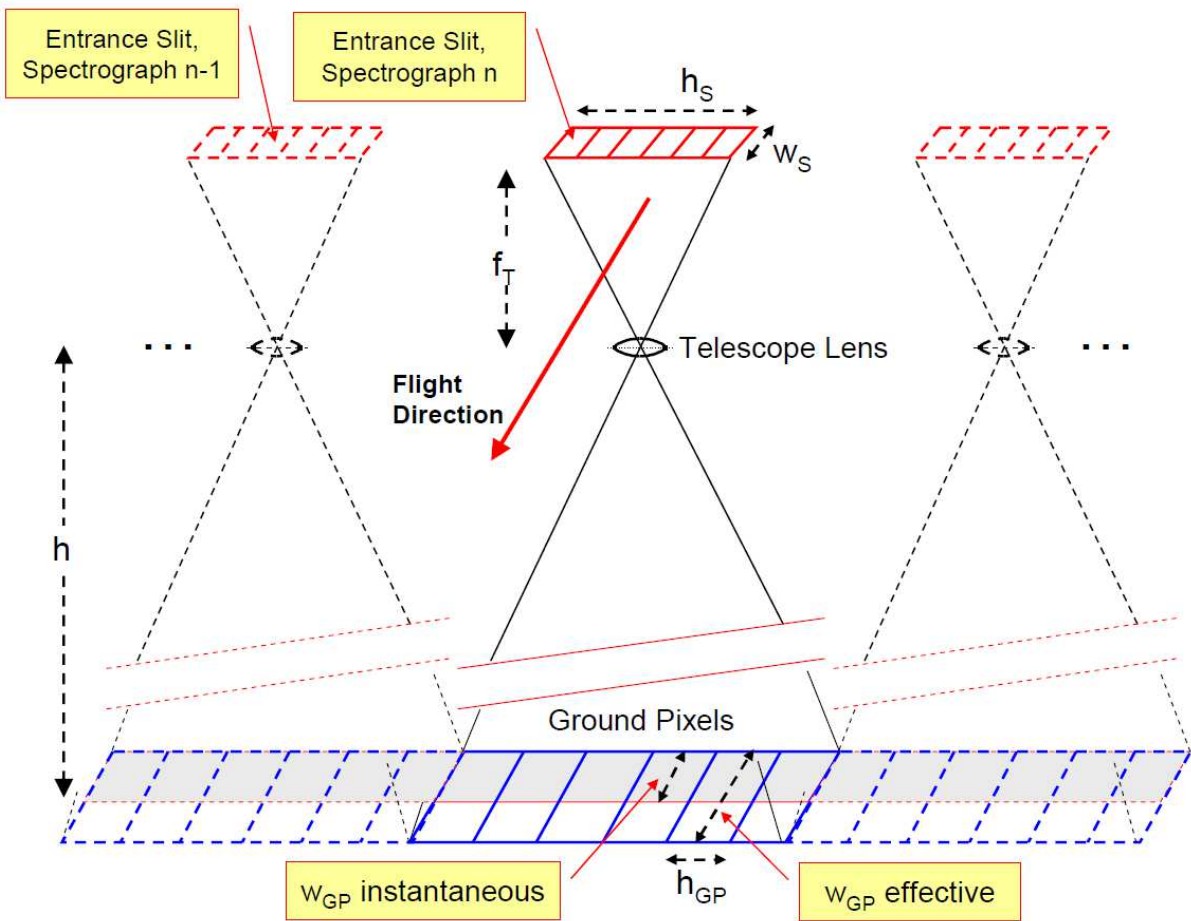

*Fig. 7b: Relationship between the spectrograph entrance slit, instantaneous field of view and the pixel dimensions. This example depicts a section of the 'scaled 1' arrangement, where each spectrograph observes 6 ground pixels, see Table 2.*

In the following we give two examples of possible satellite instruments based on arrays of micro-spectrographs. The hypothetical instrument designs are compared to the TROPOMI instrument, for simplicity we only simulated the UVvis section of TROPOMI, but other wavelength ranges could be readily added. Relevant instrument parameters are summarized in Table 2:

1) An instrument ('Scaled 1') with data similar to TROPOMI UVvis-section (Dobber et al. 2006, Veefkind et al. 2012 and Babic et al. 2019), where the individual spectrographs are scaled down to approximately 1/10. For compensation 100 spectrographs, each observing 6 ground pixels would be run in parallel. The total étendue ($\approx 0.065$ mm$^2$sr) of all spectrographs (variant A) would be somewhat smaller than the total étendue of the TROPOMI instrument ($\approx 0.103$ mm$^2$sr). Therefore, we added another variant (B) of the instrument encompassing 200 spectrometer + telescope combinations (data given in square brackets in Table 2) arranged in two identical sets of 100 spectrometer + telescope combinations.

In either case each spectrograph would have its own (now very small, see Table 2) telescope. In the case of using 200 spectrographs (variant B) each set of 6 ground pixels would be observed by two spectrographs, thus doubling the étendue (to $\approx 0.13$ mm$^2$sr)

and the signal, which would then exceed that of TROPOMI (alternatively, each spectrograph+telescope could observe only 3 ground pixels). Note that the total mass of the 'scaled 1' instrument (not just one spectrometer) as given in the last line of Table 2 is about 1/100 of that of the TROPOMI instrument. This case also illustrates the design flexibility given by the spectrograph array approach.

2) An instrument ('Scaled 2') capable of scanning at a ground pixel size of 1 km x 1 km. Here a total of 2600 spectrograph+telescope combinations would be employed, each observing 8 ground pixels, while 8 spectrographs observe the same set of 8 ground pixels. This arrangement would observe about 25-times smaller ground pixels than TROPOMI at a comparable SNR and weight.

Data for TROPOMI are taken from Veefkind et al. 2012, Kleipool et al. 2018, Babic et al. 2019, and Dobber et al. 2006. As can be seen from Table 2 scaling down the spectrograph size can provide much smaller and lighter instruments (e.g. scaled 1 will be roughly 1/100 of the weight compared to the TROPOMI instrument) while featuring similar signal to noise levels. As an option the scaled instruments could at the same time feature constant ground pixels size at the edges of the swath range while the OMI and TROPOMI instruments ground pixels are by factors of approximately 1.9 (along track) and 5.6 (cross track) larger than the nadir pixels, see for instance OMI-DUG-5.0 (2012). In Table 2 we apply a factor of up to 3.25 enhancement of magnification (i.e. enhancement of telescope focal length) in order to keep the pixel area constant across the entire swath.

The instrument (scaled 2) with 1 km by 1 km ground pixels throughout the swath with much (about 16.5) higher total étendue and comparable étendue per pixel (see Table 2) would provide a comparable signal to noise ratio as TROPOMI despite the 25-times smaller ground pixel area and could also feature constant ground pixel dimensions across the entire swath. For comparison: in order to achieve the same total étendue by just scaling up the instrument dimensions (e.g. from a TROPOMI-type instrument with $M_0 \approx 100$ kg) according to Equation 5 a total instrument mass of $\approx 67 \cdot M_0$ or around 7 metric tons would be required.

Obviously these are just examples to illustrate the potential of the new approach. Further spectrograph down-scaling (for instance to f=10mm) would be possible. In addition other combinations of spectrograph – pixel arrangements as well as the inclusion of further measures for improved light throughput (immersed gratings, imaging optics, see section 2.3) are possible and must be explored.

*Table 2: Typical data of **TROPOMI-Type and scaled** satellite instruments working in the UV for a spectral resolution of ca. 0.5 nm. Scaled 1 refers to an instrument approximately matching the data of the TROPOMI UVvis part, employing 100 (variant A) or 200 (variant B) spectrographs+telescopes, the latter values are given in square brackets. Scaled 2 refers to an instrument encompassing 2600 spectrographs+telescopes with 1 km by 1 km ground pixels.*

| Instrument<br><br>Property | TROPOMI-Type[1] | Scaled 1 | Scaled 2 |
|---|---|---|---|
| Nominal ground pixel dimensions (along track x cross track) at nadir, $km^2$ | 7 x 3.5 | 7 x 4.3 | 1 x 1 |
| Instantaneous ground pixel dimensions at nadir, (area), $km^2$ | 1.7 x 3.5 (11.9) | 1.6 x 4.3 (6.9) | 0.5 x 1 (0.5) |
| Ground pixel dimensions at edge of swath $km^2$ | 7 x 12.7 | 7 x 4.3 | 1 x 1 |
| Spectrograph focal length, mm | $\approx 200$ | 20 | 20 |
| Spectrograph F-Number | $\approx 9.5$ | 4 | 4 |
| Grating groove density, gr/mm | 2880 | 2880 | 2880 |
| Entrance slit width x height, mm x mm | $NA^2$ | 0.029 x 0.46 | 0.029 x 0.46 |
| Number of spectrographs + telescopes per instrument | 1 | 100 [200] | 2600 |
| Ground pixels per spectrograph | 576 | 6 | 8 |
| No. of spectrographs observing the same ground pixel | 1 | 1 [2] | 8 |
| Total number of ground pixels | 576 | 600 | 2600 |
| Total étendue, ($mm^2$sr) | $E_0$ ($\approx 0.103$) | $\approx 0.64 \cdot E_0$ ($\approx 0.065$) [1.27 $E_0$ (0.131)] | $\approx 16.5 \cdot E_0$ ($\approx 1.70$) |
| Étendue per pixel, $mm^2$sr | $0.000179^3$ | 0.00011 | 0.0006548 |
| Telescope focal length $f_T$ at nadir, ($f_T$ at edge of scan), mm | $NA^2$ | 14.3 ($46.5^4$) | 46.1 mm ($150^4$) |
| Telescope diameter, (dia. at the edge of scan), mm | $NA^2$ | 3.6 (11.7) | 11.5 (37.4) |
| Exposure time $\tau_{exp}$, s | 1 | 1 | 0.14 |
| Signal per pixel (signal/noise, SNR) relative to TROPOMI | 1 (1) | 0.64 (0.8) [1.3 (1.1)] | 0.51 (0.72) |
| Approximate total volume of optical system, litres[5] | 100 | ca. 0.7 [1.4] | 50-100 |
| Approximate total mass | $M_0$ ($\approx 70$ kg)[5] | $M_0$/100 [$M_0$/50] | $M_0$ |

[1]see Dobber et al. 2006, Veefkind et al. 2012, Kleipool et al. 2018, and Babic et al. 2019

[2]not applicable in this context due to intermediate imaging.

[3]calculated from telescope F-number and entrance area as given by Dobber et al. 2006 and Babic et al. 2019.

[4]a factor of up to 3.25 magnification enhancement (i.e. increase of $f_T$) is applied in order to keep the pixel area constant across the entire swath. However for 60% of the pixels (centre 1600 km of swath) the necessary extension of $f_T$ is < 2.

[5]in the case of TROPOMI only the volume of the UV/vis section

## 4.2 MAX-DOAS Applications

MAX-DOAS spectrographs are typically equipped with miniature spectrographs (e.g. Ocean Optics, Avantes). Here similar considerations apply as in the case of satellite instruments. For instance the typically used single spectrograph could be replaced by an array of scaled down spectrograph+telescope combinations as sketched in Fig. 6, In the simplest case all spectrographs could point in the same direction and the whole assembly would be tilted to measure at different elevation angles. Alternatively the spectrograph+telescopes could point at different elevations, thus analyzing the radiation at the chosen set of elevation angles simultaneously (see e.g. Leigh et al. 2007). While the latter approach would have the advantage that all elevations are observed truly simultaneously (as opposed to sequentially in the former approach), a problem could arise from slight differences in the instrument function of the individual spectrographs. Unlike the satellite case there would be no natural way where all spectrographs see the same spectrum (e.g. by observing in the zenith direction).

In either case one could argue that weight and volume of the spectrograph only constitute a small fraction of that of the entire MAX-DOAS instrument, however, the size of the instrument still scales with the spectrograph dimensions. Alternatively the scaling could be used to enhance the étendue of the instrument and thus allow proportionally faster measurements.

## 4.3 Imaging DOAS Applications

Another use of large arrays of spectrographs (+telescopes) could be imaging applications where the usually need to make a compromise between spectral-, spatial-, and temporal resolution (see e.g. Platt et al. 2015) is removed or at least relaxed. For instance an array of spectrographs (similar to the approach described by Danz et al. 2019) could arranged with a spectrograph per image pixel in a compound eye (as found in insects) fashion.

## 4.4 Other Applications

Arrays of (miniature) spectrographs could also be applied in active Long-Path DOAS (LP-DOAS) instruments. In this case a single, large telescope could be replaced by an array of small telescopes. As discussed above, the F-number of these small telescopes would be about the same as in present instruments, If the total area covered by the telescope mirrors would be the same then there would be the same light throughput as in conventional active LP-DOAS designs. In this case not only volume and weight of the spectrographs could be reduced but also the length of the telescope. This is because the F-number of each small telescope remains unchanged (compared to traditional designs), while the diameter of the mirror (or lens) – and thus its focal length f is scaled down.

## 5 SUMMARY AND CONCLUSIONS

### 5.1 Summary of Design Options

Arrays of individual (largely identical) spectrographs could help to solve a number of design challenges for both, satellite instruments and other applications.

1) Spectrograph arrays allow to improve the Étendue, and thus SNR independently from the spatial resolution.

2) Due to the scaling properties of volume and mass replacing large spectrographs by an array of smaller (identical) spectrographs can reduce the volume and mass of a spectrograph system considerably.

3) Spatial information (e.g. in satellite- or MAX-DOAS applications) could be obtained in a much simpler fashion than in present day arrangements.

4) Two-dimensional imaging detectors based on arrays of miniature spectrographs appear feasible.

Clearly, the individual spectrographs might have somewhat different responses, but this is not different to the present situation where the individual lines of pixels (corresponding to the spatial resolution) have somewhat different responses. The pointing accuracy is a matter of the platform. Minimizing possible changes between the relative pointing of the individual spectrographs is a design issue, which will not be addressed here.

## 5.2 Conclusion

We conclude that arrays of massively parallel spectrographs could solve the problem of achieving high light throughput with compact and lightweight instruments.
In particular, a reduction of the instrument volume and mass by one or two orders of magnitude at unchanged light throughput appears possible. This might be interesting for a number of particular design goals for satellite instruments:

1) Miniature satellites (e.g. Cubesats) could be equipped with spectrographs for Earth observation featuring sensitivity and spatial resolution comparable to present state of the art instruments (like GOME-2, OMI or even TROPOMI)

2) Instruments for future missions could reduce the area of the ground pixels by one or two orders of magnitude without increasing mass and size of the spectrograph.

3) If a higher mass of the instrument was allowed the spectrograph array approach allows to reduce the area of the ground pixels even further. Thus an instrument (see above) with 1 km$^2$ ground pixel size could feature comparable volume and mass of a present state of the art (e.g. the TROPOMI) spectrograph.

Also, the scaling of instruments by using the spectrograph array approach will be of great interest to other DOAS applications as well:

1) Aircraft (manned or unmanned) instruments have similar requirements as satellite instruments.
2) MAX-DOAS instruments
3) Instruments for monitoring volcanoes (as e.g. used in NOVAC)
4) Imaging DOAS applications
5) Even active LP-DOAS instruments can benefit from the Spectrograph-array approach.

We acknowledge that there might be some technical hurdles like mass production of spectrographs with as similar as possible instrument functions and other characteristics or the readout of many spectrographs in parallel. Also potential problems associated with aligning and testing many spectrographs (+telescopes) have to be solved.

In contrast to that other instrument properties, for instance radiometric accuracy and stability, initially and - in the case of satellite instruments - over the entire mission are rather questions of the particular design and have no obvious connection to the question how many individual spectrographs are used. Nevertheless, we are convinced that massively parallel miniature spectrographs are an attractive approach to future instruments.

**Acknowledgements:**

The authors like to thank the German Science Foundation (DFG) for partial funding through project PL 193/23-1-3016731.

**Author contributions:**

All authors contributed to the development of the ideas presented in the manuscript, all authors worked on aspects of the text. UP wrote most of the manuscript.

**Conflict of interest:**

The authors declare no conflict of interest.

# 6    LITERATURE

Avantes (2020), AvaSpec-Mini Data sheet, available from www.avantes.com (last accessed on Nov. 17, 2020), Avantes BV,  NL-7333 NS APELDOORN The Netherlands.

Avrutsky I., Chaganti K., Salakhutdinov I., and Auner G. (2006), Concept of a miniature optical spectrometer using integrated optical and micro-optical components, Appl. Opt. 45 (30), 7811-7817.

Babic L., Braak R., Dierssen W., Kissi-Ameyaw J., Kleipool Q., Leloux J., Loots E., Ludewig A., Rozemeijer N., Smeets J., Vacanti G., (2019), Algorithm theoretical basis document for the TROPOMI L01b data processor, The Royal Netherlands Meteorological Institute KNMI, Document No. S5P-KNMI-L01B-0009-SD, Issue 9.0.0, 2019-07-19, (https://sentinels.copernicus.eu/documents/247904/2476257/Sentinel-5P-TROPOMI-Level-1B-ATBD).

Barducci A., Guzzi D., Lastri C., Marcoionni P., Nardino V., andPippi I. (2011), Fourier Transform Spectrometry: the SNR disadvantage of the multiplex architecture, in: Imaging and Applied Optics, OSA Technical Digest (CD) (Optical Society of America, 2011), paper FWA5.DOI: 10.1364/FTS.2011.FWA5

Blumenthaler,M., Gröbner, J., Huber, M., Ambach,W., 1996. . Measuring spectral and spatial variations of UVA and UVB sky radiance. Geophys. Res. Lett . 23 (5), 547–550.

Bovensmann H., Burrows J.P., Buchwitz M., Frerick J., Rozanov V.V., Chance K.V., and Goede A.P.H. (1999), SCIAMACHY: Mission objectives and measurement modes, J. Atmos. Sci. 56, 127 – 150.

Burrows J.P., and Chance K.V. (1991), Scanning imaging absorption spectrometer for atmospheric chartography. Proc. SPIE, 1490, 146–155.

Burrows, J.P., et al. (1995), SCIAMACHY - Scanning Imaging Absorption Spectrometer for Atmospheric Chartography, Acta Astronautica, 35, 445 - 451.

Burrows J.P., (1999), The Global Ozone Monitoring Experiment (GOME): Mission concept and first scientific results, J. Atmos. Sci., 56, 151-175.

Caron J., Bézy J.-L., Courrèges-Lacoste G.B., Sierk B., Meynart R., Richert M., Loiseaux D. (2012), Polarization scramblers in Earth observing spectrometers: lessons learned from Sentinel-4 and 5 phases A/B1, International Conference on Space Optics, Ajaccio Corse, 9-12 Oct. 2012.

Czerny, M. & Turner, A.F. (1930), Über den Astigmatismus bei Spiegelspektrometern, Z. Physik 61, 792-797, https://doi.org/10.1007/BF01340206.

Chrisp M.P., Lockwood R.B., Smith M.A., Balonek G., Holtsberg C., Thome K.J., Murray K.E., and Ghuman P.(2020), Development of a compact imaging spectrometer form for the solar reflective spectral region, Appl. Optics  59 (32), 10007-10014, https://doi.org/10.1364/AO.405303.

Danz N., Höfer B., Förster E., Flügel-Paul T., Harzendorf T., Dannberg P., Leitel R., Kleinle S. and Brunner R. (2019), Miniature integrated micro-spectrometer array for snap shot multispectral sensing, Optics Express 27 (4), https://doi.org/10.1364/OE.27.005719.

Dobber M.R., Dirksen R.J., Levelt P.F., van den Oord G.H.J., Voors R.H.M., Kleipool Q., Jaross G., Kowalewski M., Hilsenrath E., Leppelmeier G.W.,de Vries J., Dierssen W., and Rozemeijer N.C. (2006), Ozone Monitoring Instrument Calibration, IEEE Trans. Geosci. Remote Sensing 44 (5), 1209.

Fastie W.G. (1952), Image Forming Properties of the Ebert Monochromator, J. Opt. Soc. Am. 42, 647-651.

Fellgett P.B. (1949), On the Ultimate Sensitivity and Practical Performance of Radiation Detectors, J. Opt. Soc. Am. 39, 970-976, doi: https://doi.org/10.1364/JOSA.39.000970.

Ferlemann F. Bauer N., Fitzenberger R., Harder H., Osterkamp H., Perner D., Platt U., Schneider M., Vradelis P., and Pfeilsticker K. (2000), Differential Optical Absorption Spectroscopy Instrument for stratospheric balloon-borne trace gas studies, Applied Optics 39, 2377-2386.

Galle B., Johansson M., Rivera C., Zhang Y., Kihlman M., Kern C., Lehmann T., Platt U., Arellano S., and Hidalgo S. (2010), Network for Observation of Volcanic and Atmospheric Change (NOVAC)— A global network for volcanic gas monitoring: Network layout and instrument description, J. Geophys. Res. 115, D05304, doi:10.1029/2009JD011823.

General S., Pöhler D., Sihler H., Bobrowski N., Frieß U., Zielcke J., Horbanski M., Shepson P., Stirm B., Simpson W., Weber K., Fischer C., and Platt U. (2014), The Heidelberg Airborne Imaging DOAS Instrument (HAIDI) A Novel Imaging DOAS Device for 2-D and 3-D Imaging of Trace Gases, J. Atmos. Meas. Tech. 7, 3459–3485, doi:10.5194/amt-7-3459-2014.

Goede A.P.H., Aarts H.J.M., van Baren C., Burrows J.P., Chance K.V., Hoekstra R., Hölzle E. Pitz W., Schneider W., Smorenburg C. Visser H., de Vries J. (1991), Sciamachy instrument design, Advances in Space Research, 11, (3), 243-246, https://doi.org/10.1016/0273-1177(91)90427-L.

Grundmann M. (2019a), Monolithic Waveguide-Based Linear Photodetector Array for Use as Ultracompact Spectrometer, IEEE Transactions on Electron Devices 66 (1), 470-477.

Grundmann M. (2019b), Modelling of a Waveguide-Based UV-Vis-IR Spectrometer Based on a Lateral (In, Ga)N Alloy Gradient, Phys. Status Solidi A 216-221, 1900170, doi: 10.1002/pssa 1900170.

Haldane J.B.S. (1927) 'On Being the Right Size' in: 'Possible Worlds and other essays', Chatto and Windus, London.

Hönninger G. and Platt U. (2002), The Role of BrO and its Vertical Distribution during Surface Ozone Depletion at Alert, Atmos. Environ. 36, 2481-2489.

Kleipool Q., Ludewig A., Babi´c L., Bartstra R., Braak R., Dierssen W., Dewitte P.-J., Kenter P., Landzaat R., Leloux J., Loots E., Meijering P., van der Plas E, Rozemeijer N, Schepers D., Schiavini D., Smeets J., Vacanti G., Vonk F., and Veefkind P. (2018), Pre-launch calibration results of the TROPOMI payload on-board the Sentinel-5 Precursor satellite, Atmos. Meas. Tech., 11, 6439–6479, https://doi.org/10.5194/amt-11-6439-2018.

Kuhn J., Bobrowski N., Wagner T., and Platt U. (2021), Mobile and high spectral resolution Fabry Pérot interferometer spectrographs for atmospheric remote sensing, Atm. Meas. Techn., preprint, https://doi.org/10.5194/amt-2021-133.

Larsson T. and Neuhaus H. (1968), The Immersion Grating: Spectroscopic Advantages and Resemblance to the Echelon Grating, Z. Naturforsch. 23 a, 2130-2132.

Leigh R.J., Corlett G.K., Frieß U., and Monks P.S. (2007), Spatially resolved measurements of nitrogen dioxide in an urban environment using concurrent multi-axis differential optical absorption spectroscopy, Atmos. Chem. Phys., 7, 4751–4762.

Levelt P.F., van den Oord G.H.J., Dobber M.R., Mälkki A., Visser H., de Vries J., Stammes P., Lundell J.O.V., and Saari H. (2006), The Ozone Monitoring Instrument, IEEE Trans. Geosci. Remote Sens., 44 (5), 1093– 1101, doi:10.1109/TGRS.2006.872333.

Meschede, Dieter (2015), Gerthsen Physik, Springer-Verlag Berlin Heidelberg, ISBN 978-3-662-45976-8.

Munro R., Lang R., Klaes D., Poli G., Retscher C., Lindstrot R., Huckle R., Lacan A., Grzegorski M., Holdak A., Kokhanovsky A., Livschitz J., and Eisinger M.(2016), The GOME-2 Instrument on the Metop Series of Satellites: Instrument Design, Calibration, and Level 1 Data Processing – an Overview, Atmos. Meas. Tech., 9, 1279–1301, doi:10.5194/amt-9-1279-2016.

OMI-DUG-5.0 (2012) Produced by OMI Team, Ozone Monitoring Instrument (OMI) Data User's Guide, January 5, 2012

Ocean Optics (2020), USB2000+ Data sheet, Ocean Optics (now: Ocean Insight), Dunedin, Florida, 34698, USA (https://www.oceaninsight.com/, last accessed on Nov. 17, 2020)

Pierson A. and Goldstein J. (1989), Stray light in spectrometers: causes and cures, Lasers and Optronics, Sept. issue, 67-74.

Platt U. and Stutz J. (2008), Differential Optical Absorption spectroscopy, Principles and Applications, XV, Springer, Heidelberg, 597 pp, 272 illus., 29 in color. (Physics of Earth and Space Environments), ISBN 978-3-540-21193-8.

Platt U., Lübcke P., Kuhn J., Bobrowski N., Prata F., Burton M.R., and Kern C. (2015), Quantitative Imaging of Volcanic Plumes – Results, Future Needs, and Future Trends, J. Volcanology Geothermal Research 300, 7-21, (JVGR, SI on Plume Imaging).

Richter A. and Wagner T. (2001), Diffuser plate spectral structures and their influence on GOME slant columns, Technical Note to ESA, January 2001,
http://joseba.mpch-mainz.mpg.de/pdf_dateien/diffuser_gome.pdf

Schilling S. (2021), Lens Array captures dim objects missed by giant telescopes, Science 371, 1301.

Sinreich R., Frieß U., Wagner T. and Platt U. (2005), Multi axis differential optical absorption spectroscopy (MAXDOAS) of gas and aerosol distributions, Faraday Discuss. 130, 153 - 164, DOI: 10.1039/B419274P.

van Amerongen A.H., Visser H., Vink R.J.P., Coppens T., Hoogeveen R.W.M. (2010), Development of immersed diffraction grating for the TROPOMI-SWIR Spectrometer, Proc. of SPIE Vol. 7826 78261D-1, doi: 10.1117/12.869018.

Veefkind J.P., Aben I., McMullan K., Förster H., de Vries J., Otter G., Claas J., Eskes H.J., de Haan J.F., Kleipool Q., van Weele M., Hasekamp O., Hoogeveen R., Landgraf J., Snel R., Tol P., Ingmann P.,
Voors R., Kruizinga B., Vink R., Visser H., and Levelt P.F. (2012), TROPOMI on the ESA Sentinel-5 Precursor: A GMES mission for global observations of the atmospheric composition for climate, air quality and ozone layer applications. Remote Sensing of Environment 120, 70–83.

Wilkes, T.C.; McGonigle, A.J.R.; Willmott, T.D.; Pering, T.D.; Cook, J.M. (2017), Low-cost 3D printed 1 nm resolution smartphone sensor-based spectrometer: Instrument design and application in
ultraviolet spectroscopy. Opt. Lett. 42 4323–4326.

Yeonjoon Park and Sang H. Choi (2013), Miniaturization of optical spectroscopes into Fresnel microspectrometers, J. of Nanophotonics 7, DOI: 10.1117/1.JNP.7.077599.

**Appendix 1:**

The change of the spectrometer with initial etendue $E_0$, initial focal length $f_0$ and optics diameter $D_0$ to $\Gamma_1 f_0$ with constant optics diameter $D_0$, can be thought of as a two step process:

1) Scale the entire spectrometer with preserved aspect ratio (according to case 1 in Table 1) by a linear factor $\Gamma_1$ (for example $\Gamma_1 = 1/2$)
→ E will be reduced to $(\Gamma_1)^2$ (i.e. to ¼ $E_0$) while the mass will change from $M_0$ to $M_0 \cdot (\Gamma_1)^3$ (i.e. to $M_0/8$). Note that the slit dimensions are also scaled by $\Gamma_1$.

2) Then increase D by factor $1/\Gamma_1$ (according to case 2a in Table 1)
→ in this step E and mass will increase by factor $1/(\Gamma_1)^2$

In total E would be unchanged, mass will be scaled to $M_0 \cdot \Gamma_1$ (i.e. to $4 \cdot M_0/8 = M_0/2$).

3) Since we assumed that in case 2b (see Table 1) the slit width is scaled, but not the slit height we have to change the slit height from $\Gamma_1 \cdot h_0$ to ist original value $h_0$.

→ The final E will be $E_0/\Gamma_1$, (i.e. $E = 2E_0$) thus $E \propto 1/M$ as given in equation 9.

