# Peer review of "The "Ideal Spectrograph" for Atmospheric Observations"

_Atmospheric Measurement Techniques, 2020_

## Referee Comment (RC2)

**1. General comments**

The paper addresses an important issue of modern Earth observation satellite missions: how to reduce the costs of the instruments while increasing their spatial resolution. As the authors rightly point out, with increased spatial resolution also the signal-to-noise of the spectrometers might need to be increased. One option to increase the signal-to-noise of a spectrometer is to increase the etendue, this is demonstrated by applying basic optic principles and by the use of volume and mass scaling of the instrument. With smaller, lighter instruments used in parallel the launch could be cheaper.

As an example to compare the new approach with, the Sentinel 5 Precursors mission with its instrument TROPOMI is used. From this example and the article's title, my expectation is that the presented approach shows to have similar or better performance for the use of atmospheric observations. However, the current content of the paper does not yet justify its title. The discussion on performance of the new design is so far mostly limited to ground pixel size and signal. Other key performance parameters, such as absolute radiometric accuracy, pointing knowledge and stability, co-registration error (deemed very important for example for the future CO2 mission, https://esamultimedia.esa.int/docs/EarthObservation/CO2M\_MRD\_v3.0\_20201001\_Issued.pdf ) are not mentioned. The proposal does not seem to include a means for solar observations to normalize the observed radiance. How can comparable radiometric accuracies be achieved in this way? These points should be addressed in an update of the manuscript.

And is the throughput of current state of the art instruments really the limiting factor for their performance? At least for TROPOMI there seems to be so much signal in some of the spectral bands that detector saturation occurs. A higher spatial resolution would in theory be possible for TROPOMI, but is – to my knowledge - limited by internal data rate restrictions.

The main assumption of the paper is that the used scaling approach for volume and mass is valid even for a large scaling factors. When looking at power or thermal lines, or shielding of detectors from cosmic radiation, there certainly is a limit to how small things can be built while keeping their functionality (and withstand the harsh space environment). Also for the spectral and radiometric stability a bigger (thermal) mass has a positive impact. As the validity of the scaling is the main assumption in the paper, a justification and the limits of the assumption should be added to the revised manuscript. For a space-borne mission power consumption is also an important factor, this should also be added.

When comparing the proposal to the example instrument, the numbers as-built should be used for the comparison, and scaled where necessary. Without the SWIR part for example, TROPOMI would be much smaller (and lighter).

As pointed out in the paper, a very good argument to use multiple spectrometers is the redundancy. On the other hand using hundreds or even thousands of devices will certainly add to the complexity of the mission. The increased complexity needs to be addressed in some way in the paper. I could imagine that there is additional data (more overhead, more housekeeping ...) to be accounted for, that (cross)calibration can be more challenging and that the combination of the data of the different spectrometers will be more complicated.

Considering how large a portion the nominal operation of a mission is of the total costs, the increased complexity in instrument control or science data processing is a trade-off which needs to be made. This needs to be made clear in the article.

To conclude, the suggestion to scale down spectrometers and use them in parallel to increase the etendue is a novel idea. As a proposal to replace or improve large high-quality space borne hyper-spectral imagers, the argumentation still shows too many gaps. I'd like to recommend to fill those taking into account the reviewers' comments and to then allow for a second review.

Section 2 of this review lists specific comments and questions about the content and understanding of the paper.

Section 3 contains a few suggestions where to edit the text to allow for a smoother read, minor spelling and grammar errors and formatting issues.

This review is based on the version amt-2020-521.pdf retrieved on the 23rd of April 2021 from https://amt.copernicus.org/preprints/amt-2020-521/. The earlier version (amt-2020-521-manuscript-version1.pdf retrieved on the 20th of January 2021) was not considered.

**2. Specific comments**

**2.1. Figures 1, 3 and 4:**

Please adapt Figures 1,3 and 4 such that it shows a properly constructed imaging path. It should also show how slit width and aperture size influence the instantaneous field of view in both along- and across-track dimension.

**2.2. Figure 7**

Please adapt Figure 7 such that it shows a more realistic scenario. At the swath width under discussion the Earth can certainly not be viewed as flat and the curvature of the Earth should be taken into account. For atmospheric retrievals also the pathlength (slant range) through the atmosphere is a point worthwhile of discussion.

**2.3. Table 2**

Please refer to the TROPOMI as-built numbers and not the design values, see for example https://sentinels.copernicus.eu/documents/247904/2476257/Sentinel-5P-TROPOMI-Level-1B-ATBD, https://doi.org/10.5194/amt-13-3561-2020 and https://doi.org/10.5194/amt-11-6439-2018

The telescope design for OMI and TROPOMI differ quite a bit, so it seems weird to compare to a mix of instruments, but title it "TROPOMI type".

- Nominal ground pixel dimension at nadir 5.5 x 3.5
- Instant. ground pixel : this is not very clear what is meant here. Do you mean the IFOV of the optics? For the TROPOMI value it should then be about 1.8 x 1.8 (so no binning applied).
- Ground pixel dimension at the edge of the swath: for TROPOMI about 9 km for across-track (no binning applied for UVIS at the edge)

- Ground pixels : as far as I know, it's less when looking at the binned nominal radiance data. Something of the order of 450. Unbinned it's around 860 pixels for the illuminated region of the detectors.
- With updated values for the focal length, F-Number (see ATBD), also the etendue needs to be recalculated
- The co-addition time is 0.84 s for the 5.5 km ground pixels
- The mass of TROPOMI is around 200 kg the total volume around 700 l. This however covers the 4 spectrographs for the four spectral regions. So if you want to restrict yourself to the UV/UVIS the smaller OMI instrument (65 kg, 70 l) would give a more realistic comparison. To make a proper comparison for mass and volume between the new proposal and the old type, the parts concerning the disregarded spectrometers, the calibration port and data handling would need to be subtracted. If you apply a mass and volume scaling here, it should be mentioned.

| #   | Page | Line      | Section | Comment                                                                                                                                                                                                                                                                                                                                                                             |
|-----|------|-----------|---------|-------------------------------------------------------------------------------------------------------------------------------------------------------------------------------------------------------------------------------------------------------------------------------------------------------------------------------------------------------------------------------------|
| SC0 | 2    | 4         | 1       | "down to 7 x 3.5 km2 (TROPOMI)", it's even 5.5 x 3.5, see
https://sentinel.esa.int/documents/247904/3541451/Sentinel-
5P-Level-1b-Product-Readme-File                                                                                                                                                                                                           |
| SC1 | 2    | 8         | 1       | "It appears clearly desirable to further shrink the ground pixel size." A justification for this statement is missing. A reference to the tracking of plumes maybe?                                                                                                                                                                                                                 |
| SC2 | 2    | 17-
19 | 1       | It's not only the shot noise adding to the noise, the read-noise
and dark current noise also needs to be taken into account. In
addition a detector pixel can only hold a certain amount of
signal before it saturates, this depends on pixel size,
technology and temperature. So the detector needs to be
chosen carefully matching throughput and read-out speed. |
| SC3 | 2    | 17-
20 | 1       | "longer exposure times $t_{exp}$ " : at least for OMI and TROPOMI
multiple exposures are co-added digitally on-board, the
number of co-additions could theoretically still be further
reduced. A single exposure needs to be long enough that the
SNR is limited by the shot noise rather than the electronic read-
out noise.                                       |
| SC4 | 5    | all       | 2.3.1   | How does the increase in entrance slit area influence the
spatial and spectral resolution? If the slit size gets larger in
along-track direction, the instantaneous field of view along-
track will get larger, or not? What is the limit for a sun-
synchronous orbit?                                                                                                 |
| SC5 | 5    | 24        | 2.3.1   | A major part of the argumentation in this paper relies on the
scaling laws for weight/size used in this paper: There are limits
where the scaling does not work that well anymore. I miss a
discussion/ a remark on the limits of scaling, see for example
Space Mission Analysis and Design: Wertz, James R., Larson.                                                  |
| SC6 | 6    | 9         | 2.3.2   | "For satellite instruments in the literature no F-numbers are
given". Please add some F-numbers, for OMI and TROPOMI,
see for example http://dx.doi.org/10.1109/TGRS.2006.869987
and
https://sentinels.copernicus.eu/documents/247904/2476257/
Sentinel-5P-TROPOMI-Level-1B-ATBD                                                                |

| #    | Page | Line | Section | Comment                                                                                                                                                                                                                                                                                                                                                                                                                                                                                                                                                                                                                                                                 |
|------|------|------|---------|-------------------------------------------------------------------------------------------------------------------------------------------------------------------------------------------------------------------------------------------------------------------------------------------------------------------------------------------------------------------------------------------------------------------------------------------------------------------------------------------------------------------------------------------------------------------------------------------------------------------------------------------------------------------------|
| SC7  | 10   | Tab1 | 2.3.5   | "No limit to scaling" at the last diffraction should be limiting, or not?                                                                                                                                                                                                                                                                                                                                                                                                                                                                                                                                                                                               |
| SC8  | 11   | All  | 3.2     | The mechanical stability is just one aspect that comes in with
the scaling. For high-quality space borne observations the
thermal stability and shielding from cosmic radiation is also
very important. Please explain the impact here.                                                                                                                                                                                                                                                                                                                                                                                                                        |
| SC9  | 11   | All  | 3.2     | At least the detectors (and possibly thermal control) will need
wiring. If more detectors are used I assume also more wiring
(and power) is needed. This is neglected in the discussion. How
does this influence the scaling? Please note that the wires
cannot easily be reduced in thickness.                                                                                                                                                                                                                                                                                                                                                             |
| SC10 | 13   | All  | 3.4     | For the amount of straylight, the distance to scattering surfaces
(optics but also surrounding mounts/walls) does play a role. I
cannot follow the argumentation that it shouldn't and that the
amount of relative straylight is the same for smaller
spectrometers. Also the separation of unwanted grating orders
is trickier if less space is available. Please provide evidence for
this statement.                                                                                                                                                                                                                                               |
| SC11 | 13   | All  | 3.6     | For the case study of this paper – individual spectrometers
covering the large swath, this section does not add anything.
However the point of combining the data on-board is a very
important point and deserves a much more detailed discussion.
In how far does the amount of data increase when using a lot
of spectrometers? What is the fraction of needed overhead
(data packaging, housekeeping, controlling) compared to the
large spectrometer case? Considering that a lot of the high
spatial resolution missions are struggling with the data volume,
this is a crucial aspect to be addressed in more detail in this
paper. |
| SC12 | 14   | AII  | 3.8     | It is great, that the authors investigate what is technically
possible at the moment. This section would certainly profit
from extending this discussion. What springs to my mind are
improvements on the grating technology (prism grating prism
combination, immersed gratings, freeform optics, use of fibre
optics). An order of magnitude of reduction in volume has
also been proposed for single spectrometer, see for example
also Crisp et al.
https://www.osapublishing.org/ao/fulltext.cfm?uri=ao-59-32-
10007&id=442323                                                                                        |
| SC13 | 14   | 30   | 4       | An order of magnitude of reduction in volume has also been proposed for single spectrometer, see for example also Crisp et al. https://www.osapublishing.org/ao/fulltext.cfm?uri=ao-59-32-10007&id=442323                                                                                                                                                                                                                                                                                                                                                                                                                                                    |
| SC14 | 15   | 25   | 4.1     | "there are a number": please be specific                                                                                                                                                                                                                                                                                                                                                                                                                                                                                                                                                                                                                                |
| SC15 | 15   | 30   | 4.1     | What do you define as separate spectrograph? A telescope +
slit + dispersive device + imaging system? Or the number of
dispersive devices with their own imaging and detector? If it is
the latter, TROPOMI has four spectrometers. If it's the former
the other numbers are not correct.                                                                                                                                                                                                                                                                                                                                                                   |

| #    | Page  | Line      | Section | Comment                                                                                                                                                                                                                                                                                                                                                                                                                                                                                                                                                                                                                                                                     |
|------|-------|-----------|---------|-----------------------------------------------------------------------------------------------------------------------------------------------------------------------------------------------------------------------------------------------------------------------------------------------------------------------------------------------------------------------------------------------------------------------------------------------------------------------------------------------------------------------------------------------------------------------------------------------------------------------------------------------------------------------------|
| SC16 | 15    | 44-
45 | 4.1     | For the missions the paper uses as reference (GOME, OMI,
TROPOMI) the instrument's alignment and the knowledge
thereof is rather critical for the mission. I would imagine that
multiplying the number of telescopes will also multiply the
need for alignment effort and calibration measurements. That
seems to be worth mentioning.                                                                                                                                                                                                                                                                                                                       |
| SC17 | 15    | 46-
48 | 4.1     | This statement is not very clear "a somewhat different function
for each viewing direction" is the reason for striping? To my
knowledge the striping is caused by subtle differences in
uncorrected residuals (for example dark current fluctuations)
when using Sun-normalized reflectance data. And the
suggested design does not seem to include a solar port, is that
correct? Also no mention is made of shutters to be able to
measure dark current.                                                                                                                                                                                             |
| SC18 | 16    | 12        | 4.1     | It would be helpful here to include that the increase of ground
pixel size towards the edges of the swath is mainly caused by
the curvature of the Earth and the resulting slanted view
towards ground. It is certainly an intriguing idea to try and
reduce this effect. I do however wonder how this would impact
the complete (gapless) coverage of the swath. To match the
along-track size a shorter co-addition time would need to
chosen, or not? If the IFOV is reduced at the edges, gaps will be
produced. And considering the slanted view, will a smaller
sampling distance indeed increase the resolution for the L2
retrievals? |
| SC19 | 17    | 23        | 4.1     | More up to date information for TROPOMI instrument
parameters can be found in:
https://sentinels.copernicus.eu/documents/247904/2476257/
Sentinel-5P-TROPOMI-Level-1B-ATBD,
https://doi.org/10.5194/amt-13-3561-2020 and
https://doi.org/10.5194/amt-11-6439-2018                                                                                                                                                                                                                                                                                                                                                                |
| SC20 | 18    | Tab 2     | 4.1     | See separate section.                                                                                                                                                                                                                                                                                                                                                                                                                                                                                                                                                                                                                                                       |
| SC21 | 19    | 1-35      | 4.2-44  | These sections do not really add information to the paper.
Please consider to omit them.                                                                                                                                                                                                                                                                                                                                                                                                                                                                                                                                                                                 |
| SC22 | 19/20 | 39-5      | 5.1     | The design challenges are at least partly addressed. What has
not been shown satisfactorily is that the arrays can also
compete with the performance of the larger instruments. What
should be discussed, are for example absolute and relative
radiometric accuracy; the achievable pointing accuracy and
knowledge; and the co-registration knowledge. The individual
spectrometers will hardly have identical response, how will this
impact the processing and combination of the data?                                                                                                                                                            |
| SC23 | 20    | 14-
16 | 5.2     | For a CUBESAT surely the date rate to downlink must be
limiting, or not? Also the attitude control is more limited than
with larger S/C. So while sensitivity and spatial resolution
might be improved, can all the data be used? Can you have
global daily coverage? What is the pointing knowledge?                                                                                                                                                                                                                                                                                                                                                           |

| #    | Page | Line      | Section | Comment                                                                                                                                                                                                                                                                                  |
|------|------|-----------|---------|------------------------------------------------------------------------------------------------------------------------------------------------------------------------------------------------------------------------------------------------------------------------------------------|
| SC24 | 20   | 19-
23 | 5.2     | Again, it also needs to be shown that the performance needed
for accurate atmospheric retrievals can be met. So not only
groundpixel size and amount of signal, but also radiometric
accuracy/stability (over the entire mission), pointing
knowledge, co-registration error |
| SC25 | 20   | 33        | 5.2     | It's good that you mention technical hurdles. The challenges of
this approach would deserve much more discussion and should
be covered in more detail earlier in the paper.                                                                                                        |

**3. Technical corrections**

The article is written in good English and easy to understand and well readable.

In the following a few minor typos and style oversights which I noticed while reading:

| #   | Page | Line | Section | Comment                                                                                                                    |
|-----|------|------|---------|----------------------------------------------------------------------------------------------------------------------------|
| TC1 | 1    | 40   | 1       | A central component of these instruments is a are moderate resolution [] grating spectrographs.                            |
| TC2 | 15   | 47   | 4.1     | This approach would have not more drawbacks, (no more means not any at all, that is not what you're trying to say I think) |
| TC3 | 18   | 8    | 4.2     | There seems to be a reference missing.                                                                                     |

---

## Author Comment (AC1)

**amt-2020-521 Answers to anonymous Referee #1:**

**We like to thank the reviewer for very carefully examining our manuscript and for making many suggestions for improvements and also for sharing a large number of practical considerations regarding the realization of a spectrograph-array with us.**

**At this point we wish to make a general statement: In our manuscript we intent to present largely theoretical considerations about new ways to significantly (i.e. by about 2 orders of magnitude) reduce volume and mass of spectrometers for environmental remote sensing applications. These considerations are based on first principles and we are glad that this is recognized by the reviewer. However, we neither intent to present a plan for actually realizing an array of spectrographs nor are our considerations restricted to satellite instruments.**

**Many of the reviewer's comments are aimed at very practical points (like the cabling, etc.), which of course will be of great important once (we hope soon) such an instrument is actually designed and manufactured. However, at the present stage, when the fundamental superiority of our approach is discussed these practical points tend to obstruct the grand view. We, therefore, answer to the points raised by the reviewer, but took the liberty to take up the majority of the technical issues raised in a general paragraph of the revised manuscript, but not in detail within other parts of the manuscript.**

**With this in mind we responded to all comments and suggestions (reproduced below in normal font) and – in most cases – made appropriate changes to the manuscript. Our responses are given in bold font below. Changes to the manuscript are given in red.**

**We are confident that we answered all questions and comments and that the revised version of the manuscript is considerably improved over the original version (in AMTD). We hope that the accordingly revised manuscript will be suitable for publication in AMT.**

**1. General comments**

The paper addresses an important issue of modern Earth observation satellite missions: how to reduce the costs of the instruments while increasing their spatial resolution. As the authors rightly point out, with increased spatial resolution also the signal-to-noise of the spectrometers might need to be increased. One option to increase the signal-to-noise of a spectrometer is to increase the étendue, this is demonstrated by applying basic optic principles and by the use of volume and mass scaling of the instrument. With smaller, lighter instruments used in parallel the launch could be cheaper.

As an example to compare the new approach with, the Sentinel 5 Precursors mission with its instrument TROPOMI is used. From this example and the article's title, my expectation is that the presented approach shows to have similar or better performance for the use of atmospheric observations. However, the current content of the paper does not yet justify its title. The discussion on performance of the new design is so far mostly limited to ground pixel size and signal. Other key performance parameters, such as absolute radiometric accuracy, pointing knowledge and stability, co-registration error (deemed very important for example for the future CO2 mission,

https://esamultimedia.esa.int/docs/EarthObservation/CO2MMRDv3.020201001Issued.pdf )
are not mentioned. The proposal does not seem to include a means for solar observations to
normalize the observed radiance. How can comparable radiometric accuracies be achieved in
this way? These points should be addressed in an update of the manuscript.

**Answer:**
**As the reviewer rightly points out the title refers to means of improving spectrometers (used
for several purposes including satellite instruments). In fact our manuscript is not only
aimed at improving Earth observation satellites. Rather, we analyze ways to dramatically
improve the light throughput of spectrometer systems of a given size and weight. Or putting
it another way, we investigate the question "how can we make spectrometer systems of a
given performance (in terms of light throughput) smaller and lighter".**
**Obviously in a given instrument other performance parameters (e.g. , pointing knowledge,
pointing stability, co-registration error, reliability, etc.) are clearly also important
(unfortunately we can not access the internal document
"CO2MMRDv3.020201001Issued.pdf"). But these parameters are mostly properties of the
platform, not properties of the spectrometer and thus they are not a topic of this
manuscript (see also our general statement).**

Changes to the manuscript: We added a paragraph at the end of the introduction more clearly
stating that we concentrate on features of the spectrograph system. In particular we make
clear that our manuscript presents largely theoretical considerations about ways to reduce
volume and mass of spectrometers for environmental remote sensing applications. These
considerations are based on first principles like well known scaling laws of nature as e.g.
spelled out by Haldane (1927) (Haldane J.B.S. (1927) 'On Being the Right Size' in: 'Possible
Worlds and other essays', Chatto and Windus, London). However, we neither intent to present
a plan for actually realizing an array of spectrographs nor are our considerations restricted to
satellite instruments. We also note there that massively parallel optics is used in other areas of
science, e.g. in astronomy (see for instance Schilling 2021) (Schilling S. (2021), Lens Array
captures dim objects missed by giant telescopes, Science 371, 1301).

**Regarding a means of direct (i.e. not reflected by Earth) solar observations to normalize the
observed radiance we note that this feature faces some technical difficulties (see discussion
of the diffuser plates) and thus is frequently not used for data evaluation. Rather 'reference
sector' techniques are employed. Overall this is rather a technological than a fundamental
point (see also our general comment).**

Changes to the manuscript: We added a sentence stating that the usual means of obtaining
direct solar spectra via diffuser plates faces some technical difficulties due to the spectral
structures introduced by the diffuser plates (see e.g. Richter A. and Wagner T., 2001, Diffuser
plate spectral structures and their influence on GOME slant columns, Technical Note to ESA,
January 2001, http://joseba.mpch-mainz.mpg.de/pdf_dateien/diffuser_gome.pdf). Therefore
the 'reference sector method' or related techniques are applied, which do not rely on disuse
plates.

And is the throughput of current state of the art instruments really the limiting factor for their
performance? At least for TROPOMI there seems to be so much signal in some of the spectral

bands that detector saturation occurs. A higher spatial resolution would in theory be possible for TROPOMI, but is – to my knowledge - limited by internal data rate restrictions.

**Answer:**
**Light throughput and thus the number of photons received in a given time interval is the ultimately limiting factor for the signal/noise ratio (SNR) attainable for nearly all spectroscopic instruments (see e.g. Platt and Stutz 2008). The fact that detector saturation may occur under certain conditions in some instruments is probably due to technical issues and by no means disproofs the fundamental relationship between Light throughput and attainable SNR (see also our general comment).**

Changes to the manuscript: We add a sentence stating that light throughput and thus the number of photons received in a given time interval is the ultimately limiting factor for the signal/noise ratio (SNR) attainable for nearly all spectroscopic instruments (see e.g. Platt and Stutz 2008).

The main assumption of the paper is that the used scaling approach for volume and mass is valid even for a large scaling factors. When looking at power or thermal lines, or shielding of detectors from cosmic radiation, there certainly is a limit to how small things can be built while keeping their functionality (and withstand the harsh space environment). Also for the spectral and radiometric stability a bigger (thermal) mass has a positive impact. As the validity of the scaling is the main assumption in the paper, a justification and the limits of the assumption should be added to the revised manuscript. For a space-borne mission power consumption is also an important factor, this should also be added.

**Answer:**
**The reviewer argues that power lines or thermal lines, or shielding of detectors from cosmic radiation may limit scaling. We think that this is rather a detail, which is mostly relevant for the actual engineering of a satellite instrument. Nevertheless, we like to explain our disagreeing view.**
**In our opinion actually the opposite is true: Smaller instruments require shorter power (or data) connection wires, also thermal ducts can be shorter and thus lighter. Similarly radiation shielding (of e.g. detectors from cosmic radiation) become much lighter for smaller instruments. Alternatively, one might want to keep the mass, and thus the power lines and shielding constant and benefit from an enhanced light throughput. Finally, it is not 'thermal mass' that matters for stability but rather thermal inertia of an instrument, which is given by its mass but also its thermal shielding (insulation). Moreover, using a large number of spectrometers in parallel has the potential advantage that the data from individual detectors, which were affected by cosmic radiation, can be sorted out and are not co-added during the evaluation procedure.**

Changes to the manuscript: We added a sentence reflecting these considerations.

When comparing the proposal to the example instrument, the numbers as-built should be used for the comparison, and scaled where necessary. Without the SWIR part for example, TROPOMI would be much smaller (and lighter).

**Answer:**
**We agree with the reviewer that our considerations as far as developed in the manuscript only pertain to the UV/vis part of TROPOMI, this is already stated in the manuscript (section 4.1, first line of Page 17 in the original manuscript). Clearly the UV/vis part of TROPOMI contributes to less than 50% to the total weight of TROPOMI.**

Changes to the manuscript: None

As pointed out in the paper, a very good argument to use multiple spectrometers is the redundancy. On the other hand using hundreds or even thousands of devices will certainly add to the complexity of the mission. The increased complexity needs to be addressed in some way in the paper. I could imagine that there is additional data (more overhead, more housekeeping ...) to be accounted for, that (cross)calibration can be more challenging and that the combination of the data of the different spectrometers will be more complicated.

Considering how large a portion the nominal operation of a mission is of the total costs, the increased complexity in instrument control or science data processing is a trade-off which needs to be made. This needs to be made clear in the article.

**Answer:**
**We agree with the reviewer to some extent. However our design also reduces complexity since a much simpler optical design can be used for each pixel (or small number of pixels), which is just repeated many times. In fact a single (or a few) rather complex instrument(s) is (are) replaced by a large number of comparatively simple instruments of identical design. The additional effort in cross calibration may be offset by less effort in the design. At present it is probably pointless to discuss this question in more detail.**

Changes to the manuscript: We added a sentence stating this fact more clearly.

To conclude, the suggestion to scale down spectrometers and use them in parallel to increase the étendue is a novel idea. As a proposal to replace or improve large high-quality space borne hyper-spectral imagers, the argumentation still shows too many gaps. I'd like to recommend to fill those taking into account the reviewers' comments and to then allow for a second review.

**Answer:**
**We are thankful to the reviewer for pointing out some potential gaps in our argumentation. But we do not quite see that these amount to „too many" gaps.**

Changes to the manuscript: See above

Section 2 of this review lists specific comments and questions about the content and understanding of the paper.

Section 3 contains a few suggestions where to edit the text to allow for a smoother read, minor spelling and grammar errors and formatting issues.
This review is based on the version amt-2020-521.pdf retrieved on the 23rd of April 2021 from https://amt.copernicus.org/preprints/amt-2020-521/. The earlier version (amt-2020-521-manuscript-version1.pdf retrieved on the 20th of January 2021) was not considered.

**2. Specific comments**

Figures 1, 3 and 4:
Please adapt Figures 1,3 and 4 such that it shows a properly constructed imaging path. It should also show how slit width and aperture size influence the instantaneous field of view in both along- and across-track dimension.

**Answer: We don't see the point of this request at this point. The Figures 1, 3, and 4 illustrate the scaling in general, they have no direct relation to the particular use of spectrometers on satellites. Regarding the request of giving more information on the instantaneous field of view in both, along- and across-track dimension we agree that this may be helpful to the reader.**

Changes to the manuscript: We add information of changes in slit size to the captions of Figures 1, 3, and 4. We add a new panel (7b) to Figure 7 relating the instantaneous pixel size to the slit dimensions.

Figure 7
Please adapt Figure 7 such that it shows a more realistic scenario. At the swath width under discussion the Earth can certainly not be viewed as flat and the curvature of the Earth should be taken into account. For atmospheric retrievals also the pathlength (slant range) through the atmosphere is a point worthwhile of discussion.

**Answer:**
**This is another point where we argue about the principle and not about details of the implementation. The principle being that our approach allows simple implementation of tailored optics, which gives constant ground pixel dimensions across the scan. We are aware of the fact that Earth is not flat, but the point of fig. 7 is to illustrate the reasons that ground pixel sizes can differ across the swath. For the sake of completeness we will change Fig. 7 to show a curved Earth surface and in the following quantify the extension of the ground pixels size towards the edges of the swath.**
**There are three reasons why the ground pixels are larger towards the edges of the swath (assumed here to be 2600 km at a satellite altitude of 800 km):**

**1) Because the pixels at the edge of the swath are further away from the instrument. This effect enlarges the pixels by a factor of 1.91 in cross-track as well as in along-track direction (area is thus enlarged by a factor of 3.64).**
**2) The pixels at the edge of the swath are seen under a larger angle ($\approx 58.4^o$) thus there cross track extension (only) is further extended by another factor of 1.91 (area is enlarged by a factor of 6.94) .**

**3) The larger angle at which the pixels at the edge of the swath are seen is further enlarged due to the curvature of Earth (by 11.5$^o$) bringing the total angle to 69.9$^o$. thus the cross track extension of the ground pixel is extended by a factor of 2.91 instead of 1.91 for the flat Earth case (or an additional factor of 1.52) The pixel area is enlarged by a factor of 10.6 over the pixel area in nadir.**
**This latter effect (curvature of Earth) has the smallest influence on the ground pixel size at the edge of the swath. Thus a ‚flat Earth' approximation could be considered for the sake of simplicity. We, nevertheless decided to follow the reviewer's suggestion and added the curvature of Earth to Fig. 7.**

Changes to the manuscript: We added text to more clearly to describe the situation as pointed out in the answer to the reviewer and changed Fig. 7 (now Fig. 7a) and its caption adding the information detailed above. We also changed the required dimensions of the telescopes at the edge of scan in Table 2 so that the pixel area at the edge of the scan remains the same as in nadir (this requires an extension of the focal length by a factor of 3.25 at the edge of scan over the focal length of the nadir pointing telescopes).

Table 2
Please refer to the TROPOMI as-built numbers and not the design values, see for example https://sentinels.copernicus.eu/documents/247904/2476257/Sentinel-5P-TROPOMI-Level-1B-ATBD, https://doi.org/10.5194/amt-13-3561-2020 and https://doi.org/10.5194/amt-11-6439-2018
The telescope design for OMI and TROPOMI differ quite a bit, so it seems weird to compare to a mix of instruments, but title it "TROPOMI type".
•        • Nominal ground pixel dimension at nadir 5.5 x 3.5

•        • Instant. ground pixel : this is not very clear what is meant here. Do you mean the

IFOV of the optics? For the TROPOMI value it should then be about 1.8 x 1.8 (so no binning applied).

•        • Ground pixel dimension at the edge of the swath: for TROPOMI about 9 km for across-track (no binning applied for UVIS at the edge)

**Answer: We like to thank the reviewer for reminding us of the available information on TROPOMI. We are aware of the fact that OMI and TROPOMI are different instruments, but the reviewer will admit that there are, nevertheless, similarities (2D-detectors, similar orbits, similar overall design of the optics).**

Changes to the manuscript: We added references to the "Algorithm theoretical basis document for the TROPOMI L01b data processor" (https://sentinels.copernicus.eu/documents/247904/2476257/Sentinel-5P-TROPOMI-Level-1B-ATBD)" and to Kleipool et al. 2018 (Pre-launch calibration results of the TROPOMI payload on-board the Sentinel-5 Precursor satellite, Atmos. Meas. Tech. 11, 6439–6479, https://doi.org/10.5194/amt-11-6439-2018).

•        • Ground pixels : as far as I know, it's less when looking at the binned nominal radiance data. Something of the order of 450. Unbinned it's around 860 pixels for the illuminated region of the detectors.

• • With updated values for the focal length, F-Number (see ATBD), also the étendue needs to be recalculated

• • The co-addition time is 0.84 s for the 5.5 km ground pixels

• • The mass of TROPOMI is around 200 kg the total volume around 700 l. This however covers the 4 spectrographs for the four spectral regions. So if you want to restrict yourself to the UV/UVIS the smaller OMI instrument (65 kg, 70 l) would give a more realistic comparison. To make a proper comparison for mass and volume between the new proposal and the old type, the parts concerning the disregarded spectrometers, the calibration port and data handling would need to be subtracted. If you apply a mass and volume scaling here, it should be mentioned.

**Answer: We like to thank the reviewer for the information on TROPOMI and OMI. We will change the relevant data in Table 2 as suggested. As for the weight of the UV/UVVIS section of TROPOMI or the total weight of OMI we assumed a weight of 100kg ($M_0$, see Table 2), which is not far from the figure estimated by the reviewer. We also like to point out that the scaling laws are independent of the initial mass. The question of the scaling of data handling is a rather technical point again (see our general comment). In the case of electronics it is clear the size and mass fraction of the electronics is diminishing due to the progress in technology.**

Changes to the manuscript: We added a sentence explaining more clearly that our comparison only pertains to the UV or UVVIS section of TROPOMI.

| # | Page | Line | Section | Comment |
|---|---|---|---|---|
| SC0 | 2 | 4 | 1 | "down to 7 x 3.5 km2 (TROPOMI)" , it's even 5.5 x 3.5, see https://sentinel.esa.int/documents/247904/3541451/Sentinel-5P-Level-1b-Product-Readme-File

**Answer: We like to thank the reviewer for this information** |
| SC1 | 2 | 8 | 1 | "It appears clearly desirable to further shrink the ground pixel size." A justification for this statement is missing. A reference to the tracking of plumes maybe?

**Answer: There are a number of obvious reasons why smaller ground pixels are desirable:**
**1) Smaller structures like volcanic plumes, emissions from individual sources, ship tracks, etc. could be better resolved or resolved at all.**
**2) The sensitivity for small structures is improved (ideally inversely to the square of the pixel linear dimension), since the signal is not 'smeared out'**
**3) …** |
| SC2 | 2 | 17-19 | 1 | It's not only the shot noise adding to the noise, the read-noise and dark current noise also needs to be taken into account. In addition a detector pixel can only hold a certain amount of signal before it saturates, this depends on pixel size, technology and temperature. |

So the detector needs to be chosen carefully matching throughput and read-out speed.

**Answer: In modern detectors read-out noise and dark current noise are negligible compared to the photon shot noise. Clearly, the detectors and its operating conditions have to be chosen that the detector does not saturate. In this context smaller ground pixels also mean shorter readout times (as shown in Table 2) thus making it easier to avoid detector saturation.**

Change to the manuscript: We added text to section 4.1 explaining these facts.

| SC3 | 2 | 17-20 | 1 | "longer exposure times $t_{exp}$" : at least for OMI and TROPOMI multiple exposures are co-added digitally on-board, the number of co-additions could theoretically still be further reduced. A single exposure needs to be long enough that the SNR is limited by the shot noise rather than the electronic read-out noise. |
|---|---|---|---|---|
| | | | | **Answer: See answer to SC2** |
| SC4 | 5 | all | 2.3.1 | How does the increase in entrance slit area influence the spatial and spectral resolution? If the slit size gets larger in along-track direction, the instantaneous field of view along-track will get larger, or not? What is the limit for a sun-synchronous orbit? |
| | | | | **Answer: The reviewer is correct here, larger (wider) slit means larger instantaneous field of view in along-track direction. Data are given in Table 2** |
| | | | | Change to the manuscript: We added text to section 4.1 stating this relationship. |
| SC5 | 5 | 24 | 2.3.1 | A major part of the argumentation in this paper relies on the scaling laws for weight/size used in this paper: There are limits where the scaling does not work that well anymore. I miss a discussion/ a remark on the limits of scaling, see for example Space Mission Analysis and Design: Wertz, James R., Larson. |
| | | | | **Answer: We are a bit surprised by this comment. In sections 3.2 ("Is it true that the spectrograph mass scales with $L^3$") and 3.3 ("How far can we shrink a spectrograph") we discuss exactly these points.** |
| SC6 | 6 | 9 | 2.3.2 | "For satellite instruments in the literature no F-numbers are given". Please add some F-numbers, for OMI and TROPOMI, see for example http://dx.doi.org/10.1109/TGRS.2006.869987 and https://sentinels.copernicus.eu/documents/247904/2476257/Sentinel-5P-TROPOMI-Level-1B-ATBD |
| | | | | **Answer: We like to thank the reviewer for this information** |
| | | | | Change to the manuscript: We added the F-numbers |

| # | Page | Line | Section | Comment |
|---|------|------|---------|---------|
| SC7 | 10 | Tab1 | 2.3.5 | "No limit to scaling" at the last diffraction should be limiting, or not?

**Answer: Certainly, diffraction is limiting scaling as discussed in section … However, here we discuss the limits of scaling up light throughput.**

Changes to the manuscript: We clarify the meaning of scaling here in order to avoid misunderstanding. |
| SC8 | 11 | All | 3.2 | The mechanical stability is just one aspect that comes in with the scaling. For high-quality space borne observations the thermal stability and shielding from cosmic radiation is also very important. Please explain the impact here.

**Answer: See above …**
**We note that thermal stability is given by the ratio of heat flow, which is proportional to the surface of an object, and its heat capacity, which is (for a given material) proportional to the volume of the object. Thus the thermal time constant of a spectrometer should scale with $1/L$.**

Changes to the manuscript: We add an explanation. One might as well keep the original mass and thermal time constant and benefit from an enhanced light throughput. |
| SC9 | 11 | All | 3.2 | At least the detectors (and possibly thermal control) will need wiring. If more detectors are used I assume also more wiring (and power) is needed. This is neglected in the discussion. How does this influence the scaling? Please note that the wires cannot easily be reduced in thickness.

**Answer: This is an interesting point. As pointed out above it can be assumed that a smaller instrument requires shorter and thus lighter wiring (which amounts to only a small fraction of the total weight anyway). Moreover modern electronics would use a bus design for attaching detectors to a central electronics box, which should further reduce the fraction of weight devoted to wiring.**

Changes to the manuscript: We add a sentence explaining this relationship. |
| SC10 | 13 | All | 3.4 | For the amount of straylight, the distance to scattering surfaces (optics but also surrounding mounts/walls) does play a role. I cannot follow the argumentation that it shouldn't and that the amount of relative straylight is the same for smaller spectrometers. Also the separation of unwanted grating orders is trickier if less space is available. Please provide evidence for this statement. |

**Answer: The amount of straylight is proportional to ratio of the area of the scattering surfaces and the detector area, i.e. basically it scales with the inverse of the F-Number. However, we assume the f-number to be kept constant while scaling. So there should be no change in stray light.**

Changes to the manuscript: We add text explaining this relationship.

| SC11 | 13 | All | 3.6 | For the case study of this paper – individual spectrometers covering the large swath, this section does not add anything. However the point of combining the data on-board is a very important point and deserves a much more detailed discussion. In how far does the amount of data increase when using a lot of spectrometers? What is the fraction of needed overhead (data packaging, housekeeping, controlling...) compared to the large spectrometer case? Considering that a lot of the high spatial resolution missions are struggling with the data volume, this is a crucial aspect to be addressed in more detail in this paper. |

**Answer: The idea is to basically have one (or a small number of) spectrometer(s) per viewing direction. In the case of one spectrometer (with 1-dimensional detector) per viewing direction there would be no change in the amount of data generated compared to an approach using a spectrograph with 2-dimensional detector (as in OMI or TROPOMI). Of course a larger number of ground pixels (as in the case of 1km x 1km pixels) would generate proportionally more data. In designs where several spectrometers observe the same ground pixel the data could be co-added on board. Thus, again there would be no increase in data rate compared to a single large-spectrometer - 2-dim. approach (like OMI or TROPOMI).**
**Consequently the fraction needed for overhead (data packaging, housekeeping, controlling, etc. ) would also be unchanged.**

Changes to the manuscript: We add text explaining this relationship.

| SC12 | 14 | All | 3.8 | It is great, that the authors investigate what is technically possible at the moment. This section would certainly profit from extending this discussion. What springs to my mind are improvements on the grating technology (prism grating prism combination, immersed gratings, freeform optics, use of fibre optics...). An order of magnitude of reduction in volume has also been proposed for single spectrometer, see for example also Crisp et al. https://www.osapublishing.org/ao/fulltext.cfm?uri=ao-59-32-10007&id=442323 |

**Answer: We feel commended by the reviewer. We shall include a reference to Crisp et al. 2020 as an example of 'conventional' improvements i.e. by better throughput or optimized optics. Although the spectrometer described by these authors has a**

**rather poor spectral resolution of several 10nm rendering it unusable for most spectroscopic applications.**

Changes to the manuscript: We add a reference to Crisp et al. 2020 as an example of 'conventional' improvements

| SC13 | 14 | 30 | 4 | An order of magnitude of reduction in volume has also been proposed for single spectrometer, see for example also Crisp et al. https://www.osapublishing.org/ao/fulltext.cfm?uri=ao-59-32-10007&id=442323 |
|------|----|----|---|---|

**Answer:  see answer to preceding comment.**

Changes to the manuscript: We add a reference to Crisp et al. 2020 as an example of 'conventional' improvements

| SC14 | 15 | 25 | 4.1 | "there are a number": please be specific |
|------|----|----|-----|---|

**Answer: We see no point in counting the instruments (e.g. GOME, GOME-2A, B, C, SCIAMACHY, ODIN, OMI, TROPOMI, OMPS NM, GEMS (geostationary), EMI )**

| SC15 | 15 | 30 | 4.1 | What do you define as separate spectrograph? A telescope + slit + dispersive device + imaging system? Or the number of dispersive devices with their own imaging and detector? If it is the latter, TROPOMI has four spectrometers. If it's the former the other numbers are not correct. |
|------|----|----|-----|---|

**Answer:  We largely mean the latter (although in our proposal each spectrometer has its own telescope), but refer to the UV/vis part of TROPOMI only.**

Changes to the manuscript: We add text to make this point more clear.

| # | Page | Line | Section | Comment |
|---|------|------|---------|---------|
| SC16 | 15 | 44-45 | 4.1 | For the missions the paper uses as reference (GOME, OMI, TROPOMI) the instrument's alignment and the knowledge thereof is rather critical for the mission. I would imagine that multiplying the number of telescopes will also multiply the need for alignment effort and calibration measurements. That seems to be worth mentioning. |

**Answer:  There should be no substantial change in the need for calibration measurements since all spectrometers can be calibrated simultaneously, just like all spatial channels of a 2-dim. imaging spectrometer are. The multiple telescopes would need some kind of alignment to several tenth of a degree or somewhat less than a tenth of a degree in the TROPOMI-equivalent or 1km resolution, respectively. This would be an extra effort, however it is likely that the alignment could be**

**achieved by precise machining of the base plate.**

| | | | | |
|---|---|---|---|---|
| SC17 | 15 | 46-48 | 4.1 | This statement is not very clear "a somewhat different function for each viewing direction" is the reason for striping? To my knowledge the striping is caused by subtle differences in uncorrected residuals (for example dark current fluctuations) when using Sun-normalized reflectance data. And the suggested design does not seem to include a solar port, is that correct? Also no mention is made of shutters to be able to measure dark current. |

**Answer: We agree with the reviewer that the reason for the striping is not fully understood. However, from experiences with OMI and TROPOMI, there are indications that the dominant source of striping and its correction are a problem that is similar to effects of a "somewhat different instrument functions for each viewing direction" or possibly other subtle differences between individual instruments. Again, we feel that this is not a fundamental, but rather technical point (see also our general comment).**

Changes to the manuscript: We added a sentence to make this point more clear.

| | | | | |
|---|---|---|---|---|
| SC18 | 16 | 12 | 4.1 | It would be helpful here to include that the increase of ground pixel size towards the edges of the swath is mainly caused by the curvature of the Earth and the resulting slanted view towards ground. It is certainly an intriguing idea to try and reduce this effect. I do however wonder how this would impact the complete (gapless) coverage of the swath. To match the along-track size a shorter co-addition time would need to chosen, or not? If the IFOV is reduced at the edges, gaps will be produced. And considering the slanted view, will a smaller sampling distance indeed increase the resolution for the L2 retrievals? |

**Answer:  see above**

| | | | | |
|---|---|---|---|---|
| SC19 | 17 | 23 | 4.1 | More up to date information for TROPOMI instrument parameters can be found in: https://sentinels.copernicus.eu/documents/247904/2476257/Sentinel-5P-TROPOMI-Level-1B-ATBD, https://doi.org/10.5194/amt-13-3561-2020 and https://doi.org/10.5194/amt-11-6439-2018 |

**Answer:  Unfortunately this link does not appear to work**

| | | | | |
|---|---|---|---|---|
| SC20 | 18 | Tab 2 | 4.1 | See separate section. |
| SC21 | 19 | 1-35 | 4.2-44 | These sections do not really add information to the paper. Please consider to omit them. |

**Answer: We disagree with the reviewer here. It appears to be a misunderstanding on the part of the reviewer that this manuscript is solely about satellite instruments. As pointed out**

**in several places of the manuscript satellites are only one application. Therefore we consider it very important to give at least some amount of information on other possible application of spectrograph arrays.**

Changes to the manuscript: none.

| | | | | |
|---|---|---|---|---|
| SC22 | 19/20 | 39-5 | 5.1 | The design challenges are at least partly addressed. What has not been shown satisfactorily is that the arrays can also compete with the performance of the larger instruments. What should be discussed, are for example absolute and relative radiometric accuracy; the achievable pointing accuracy and knowledge; and the co-registration knowledge. The individual spectrometers will hardly have identical response, how will this impact the processing and combination of the data? |

**Answer: This is a point that refers to very practical issues and are thus – as pointed out above - not the topic of this manuscript. Clearly the individual spectrometers might have somewhat different responses, but this is not different to the present situation where the individual lines of pixels (corresponding to the spatial resolution) have somewhat different responses. The pointing accuracy is a matter of the platform. Minimizing possible changes between the relative pointing of the individual spectrographs is a design issue, which will not be addressed in this manuscript.**

Changes to the manuscript: We add text to explain this point.

| | | | | |
|---|---|---|---|---|
| SC23 | 20 | 14-16 | 5.2 | For a CUBESAT surely the date rate to downlink must be limiting, or not? Also the attitude control is more limited than with larger S/C. So while sensitivity and spatial resolution might be improved, can all the data be used? Can you have global daily coverage? What is the pointing knowledge? |

**Answer: We consider this a rather rhetorical question. Obviously a Cubesat-type platform - based imaging spectrometer might not have the 1km ground pixel resolution. However, our point is that an instrument matching the ground pixel size of TROPOSAT-type instruments when using arrays of scaled down spectrographs. Cubesat-type microsatellites undergo a rapid evolution, for instance due to magnetorquers (Niccolò Bellini (2014), Magnetic actuators for nanosatellite attitude control, thesis, University of Bolbogna, Italy (https://amslaurea.unibo.it/7506/1/Bellini_Niccolò_Tesi.pdf).) they can have relatively good orientation control.**

| # | Page | Line | Section | Comment |
|---|------|------|---------|---------|
| SC24 | 20 | 19-23 | 5.2 | Again, it also needs to be shown that the performance needed for accurate atmospheric retrievals can be met. So not only ground pixel size and amount of signal, but also radiometric accuracy/stability (over the entire mission), pointing knowledge, co-registration error.... |

**Answer: We agree with the reviewer in principle, however we see little connection between using a different approach to spectrometer design and questions like pointing knowledge or co-registration errors. Also, radiometric accuracy and stability, initially and over the entire mission are rather questions of the particular design and have no obvious connection to the question how many individual spectrographs are used. We also like to refer to our statement regarding fundamental vs. practical questions at the beginning of this document.**

Changes to the manuscript: We add some text to explain this point.

| # | Page | Line | Section | Comment |
|---|------|------|---------|---------|
| SC25 | 20 | 33 | 5.2 | It's good that you mention technical hurdles. The challenges of this approach would deserve much more discussion and should be covered in more detail earlier in the paper. |

**Answer: Again we also like to refer to our statement regarding fundamental vs. practical questions at the beginning of this document. We, therefore, disagree with the reviewer that "The challenges of this approach would deserve much more discussion "In fact we provide quite some discussion about technical challenges in the original version of the manuscript and added some more in response to the reviewers' comments. Therefore we feel that more discussion would rather obscure the manuscript than adding more clarity. Of course we recognize that once such an instrument is really build – like always - many minor (and possibly major) technical hurdles have to be overcome.**

Changes to the manuscript: None at this point.

**3. Technical corrections**

The article is written in good English and easy to understand and well readable.

In the following a few minor typos and style oversights which I noticed while reading:

| # | Page | Line | Section | Comment |
|---|------|------|---------|---------|
| TC1 | 1 | 40 | 1 | A central component of these instruments is a are moderate resolution [...] grating spectrographs. |
| TC2 | 15 | 47 | 4.1 | This approach would have not more drawbacks, (no more means not any at all, that is not what you're trying to say I think) |
| TC3 | 18 | 8 | 4.2 | There seems to be a reference missing. |

**Answer: we made the suggested changes and added the missing reference on page 19 (not 18): Leigh R.J., Corlett G.K., Frieß U., and Monks P.S. (2007), Spatially resolved measurements of nitrogen dioxide in an urban environment using concurrent multi-axis differential optical absorption spectroscopy, Atmos. Chem. Phys., 7, 4751–4762.**

---

## Author Comment (AC2)

**Answers to Anonymous Referee #2, 29 Apr 2021**

'Number of comments on amt-2020-521.'

**We like to thank the reviewer for carefully examining our manuscript and for making many suggestions for improvements.**

**At this point we wish to make a general statement: In our manuscript we intent to present largely theoretical considerations about new ways to significantly (by two orders of magnitude) reduce volume and mass of spectrometers for environmental remote sensing applications. These considerations are based on first principles and we are glad that this is recognized by the reviewer. However, we neither intent to present a plan for actually realizing an array of spectrographs nor are our considerations restricted to satellite instruments.**

**Several of the reviewer's comments are aimed at very practical points (like the cabling, etc.), which of course will be of great important once (we hope soon) such an instrument is actually designed and manufactured. However, at the present stage, when the fundamental superiority of our approach is discussed these practical points tend to obstruct the grand view. We, therefore, answer to the points raised by the reviewer, but took the liberty to take up the majority of the technical issues raised in a general paragraph of the revised manuscript, but not in detail within other parts of the manuscript.**

**With this in mind we responded to all comments and suggestions (reproduced below in normal font) and – in most cases – made appropriate changes to the manuscript. Our responses are given in bold font below. Changes to the manuscript are given in red.**

**We are confident that we answered all questions and comments and that the revised version of the manuscript is considerably improved over the original version (in AMTD). We trust that the accordingly revised manuscript will be suitable for publication in AMT.**

Reviewer's comment (here and throughout the rest of the document in normal font):
This is an interesting paper for exploring and discussing new possibilities to perform atmospheric trace gas measurements using satellite-based spectrographs that try to fundamentally improve on the concepts of predecessor instruments.

**Answer: We like to thank the reviewer fort he positive assessment of our manuscript.**

Traditionally, the sizing of predecessor instrument has been driven by user requirements on signal-to-noise and spectral resolution and sampling that are in some cases questionable when the resulting L2 data products are inspected afterwards.

**Answer: Not quite clear what the reviewer is referring to.**

The paper could benefit from a section of requirements in order to clarify which L2 data products are targeted with the optimised spectrometer design.

**Answer: The requirements for an instrument based on our proposed design principles would be no different from the requirements for e.g. TROPOMI. We simply suggest a design that would accomplish the same with about one hundredth of the mass.**

For example, from use perspective, the high spatial resolution of e.g. 1 km x 1 km in the spectral range 270-500 nm would 'only' benefit NO2 retrievals, for which a very limited spectral range (e.g. 420-450 nm) with high spectral resolution would be sufficient.

**Answer: In our opinion from a high spatial resolution of e.g. 1 km x 1 km in the spectral range 270-500 nm not only the $NO_2$ retrievals would benefit, but also the retrieval of $SO_2$, BrO, IO, OClO, HCHO, $O_4$, glyoxal, and water vapour.**

Changes to the manuscript: We add text clarify that point.

It is the combination of various requirements for various L2 products that are all given priority 1 that often drive the size and mass of these type of satellite UV-VIS-NIR-SWIR spectrometers.

**Answer: See above**

Properly accounting for polarisation has had a tendency in the past to increase instrument size (see also section 3.5).

Accepting the resulting errors from ignoring polarisation can be used to reduce the instrument size (and mass) considerably.

**Answer: We agree with the reviewer regarding the question whether the polarisation measurement should be made or whether accepting the resulting errors from ignoring polarisation should not rather be used to reduce the instrument size (and mass).**

Changes to the manuscript: We add text clarify that point.

In my view the paper could be further improved with some additional comparison information:

- Size comparison for predecessor instruments.
- Size comparison for 1D vs 2D predecessor instruments.

**Answer: While we agree with the reviewer that more size comparisons may be interesting in principle we state that the aim of the manuscript is not the comparison of satellite instruments (that would be an interesting task in itself) but to propose a new approach to**

**spectroscopic instrumentation for environmental measurements – from satellite, other platforms, or from the ground. The size comparisons we are giving should be seen as examples fort he new possibilities.**

Changes to the manuscript: We add text clarify this point.

Section 3.5 Further considerations

I don't share this statement:

"Obviously, for very small spectrometers the depolarizer will also be very small, thus adding negligibly to the volume and weight of the instrument."

Depolarizer plates work on the basis of spatial randomisation of the atmospheric polarization, which requires a certain minimal spatial size of the entrance aperture and depolarization plates. Has this been considered?

**Answer: The reviewer gives no argument why the depolarizer should not be very small. In the following we will give a brief argument why our statement regarding the small (compared to the spectrometer+telescope) size of a polarisation scrambler is likely to be valid:**
**In order to estimate the size of a possible polarisation scrambler we have a look at the sketch of a depolarizer in Fig. 1. Across the combination of two wedged plates (shown in cross section) the polarisation changes from: Linear (e.g. vertical as shown in Fig. 1) - circular – linear (90$^o$ twisted) – circular (opposite direction) – linear (vertical again). The required change of thickness $\Delta d$ across the plate is given by $\Delta d = \lambda/\Delta n$, where $\Delta n$ denotes the difference in the index of refraction for the ordinary ray and extraordinary ray, respectively. Typical values of $\Delta n$ are around 0.01 (e.g. 0.009 for quartz). For $\lambda$=400nm a $\Delta d$ of about 40$\mu$m would result. The polarisation scrambler would e.g. be mounted in front of the telescope. At a telescope diameter $D_T$ of e.g. 12mm (see Table2, scaled 2-instrument) this would result in a wedge angle of $\alpha\approx$0.003 radian or about $\approx$0.2 degrees. In practice one might prefer not have only one cycle through the polarisation states, but rather e.g. 10 cycles, corresponding to an angle $\alpha$ of $\approx$2 degrees, (which is in fact close to the wedge angles of 0.6$^o$ and 1.35$^o$ given for the TROPOMI polarisation scrambler, as given in the "Algorithm theoretical basis document for the TROPOMI L01b data processor", document number S5P-KNMI-L01B-0009-SD, CI identification : CI-6480-ATBD, issue 8.0.0 of June 1, 2017). The plate could be 1-2mm thick. In summary, this small device would hardly contribute much to the total weight or volume of an individual spectrometer+telescope assembly. Regarding the comment of the reviewer that a "certain minimal spatial size of the entrance aperture and depolarization plates" is required, judging from the TROPOMI telescope size (25mm$^2$ area as stated in the "Algorithm theoretical basis document …", corresponding to a diameter <6mm) the scrambler diameter of the proposed 'scaled 2' instrument would be twice as large than that of TROPOMI.**

Changes to the manuscript: We added a sentence stating that the size of a possible polarisation scrambler would be comparable to the device used in TROPOMI and its weight and volume would be small compared to the spectrometer+telescope units.

**Fig. 1: Cross-section through a polarisation scrambler: Two wedges, the left made from birefringent material (e.g. crystalline quartz), the right from non-birefringent material (e.g. vitreous quartz) act to change and twist the polarisation state (shown in blue). The non-birefringent wedge is only needed for compensation. By superimposing many polarisation states the original degree of polarisation of the incoming radiation polarisation is (nearly) lost.**

Section 3.6 How to combine the signal of a large number of spectrographs?

This chapter is a bit an oversimplification of a post-processing issue to combine the data of the various spectrometers that may become problematic in terms of efforts and processing power.

**Answer: Clearly, there will be some software development needed, however it is about e.g. shifting/stretching spectra to match each other, well known principles (see e.g. Platt and Stutz 2008)**

Changes to the manuscript: We add text to clarify that point.

Section 3.7 How to manufacture arrays of (micro) spectrographs?

This is a bit of an oversimplification, because the issue is not so much the manufacturing costs, but more the space qualification and documentation costs.

**Answer:  It is likely that some development will be needed. However, it is unclear whether space qualification and documentation of a large number of identical (and rather simple) spectrographs is more effort than that of a single (relatively complicated) spectrograph.**

Changes to the manuscript: We add a sentence to make that point.

Section 4.1 is not fully understood, because here the authors start to use two competing objectives without making clear (to the opinion of the reviewer) how these are combined:

1.  Using multiple small spectrographs to each observe a different ground pixel, whereas an instrument such as TROPOMI observes all ground pixels simultaneously.

2.  Using multiple small spectrographs to each observe the same ground pixel, in order to improve signal and signal to noise using multiple similar small spectrographs instead of one bigger one.

For example, in table 2, the TROPOMI type has 576 ground pixels per spectrograph, whereas the 'Scaled 1' has 6 ground pixels per spectrograph.

Hence to cover the same spatial range and resolution on the ground the 'Scaled 1' needs 576/6=96 spectrographs.

The number of spectrographs difference is 100, which thus covers only for the above spatial range/resolution per spectrograph, not for signal / signal-to-noise.

**Answer: In section 4.1 we explain that there could be one spectrograph per viewing direction or several („one or more"). These two approaches are not competing, they are just options in case the light throughput of a single, scaled down spectrograph is not sufficient then several spectrographs (with their own telescope each) could observe the same ground pixel.**

Changes to the manuscript: We add text clarify that particular option.

Using mechanical scanners, as mentioned earlier in the section, doesn't comply with the simplicity and small design of the downscaled spectrographs.

**Answer:  This is a misunderstanding, in line 33 of page 15 (which is the only point in this section where we mention a scanner) we refer to scanners being used by existing, conventional instruments (e.g. the GOME-series). We do not suggest to use mechanical scanners.**

Changes to the manuscript: We add text clarify that point.

I find the conclusion section 5.2 somewhat biased and not considering all advantages and disadvantages equally, and to some extent also comparing different things. See also the above comments.

**Answer: Obviously our summary is trying to list the strong points, but we also point out some problems at the end of the summary. In order to come to a more balanced presentation we added text to list more possible hurdles to the implementation as given by both reviewers.**

Changes to the manuscript: We added text to list more possible hurdles to the implementation as given by both reviewers.

I am not convinced that at the same spatial resolution and range, same spectral resolution and range and the same user / L2 requirements the option of the array of identical downscaled spectrographs presents a significant mass volume advantage. I find this aspect is not conclusively demonstrated in the preceding sections and I encourage the authors to improve this demonstration, including the aspects also mentioned here, e.g. by clearly separating the aspects of number of ground pixels per spectrograph and signal/signal-to-noise per spectrograph.

**Answer: We are somewhat surprised by this conclusion of the reviewer, since we demonstrate the advantage of arrays of small spectrographs at length in section 3. The further questions regarding the possible confusion of one or several spectrographs (+telescope) observing the same ground pixel were answered above (question to section 4.1).**

I also recommend that clear recommendations for the downscaled spectrographs for spectral range/resolution and the use of two-dimensional detector arrays vs. the use of mechanical scanners are given, because the use of scanners in small spectrographs is not an option and the use of two-dimensional arrays complicates the optics (as pointed out in the paper).

**Answer: We do not recommend mechanical scanners (see answer to the mechanical scanner issue, above. Whether mechanical scanners in conjunction with miniaturized spectrographs make sense may be a target of future investigations.**

Changes to the manuscript: None, since the point was taken up above.

In addition, operating a fleet of small satellites with small spectrographs is an expensive undertaking. Which operational institution (or company) is supposed to do this and at what cost?

**Answer: We think that this is a question that goes beyond the scope of the manuscript. Whether a fleet of small satellites (how many?) would really be more expensive than a large single satellite is to be found out. Clearly, the idea of Cubesat and similar micro**

**satellites is to make them cheap. Regarding the question who could finance and operate such a fleet: We presently see a surge of private space activities. Thus it is possible (and even likely) that these NGOs will be interested in Earth observation in the near future.**

Changes to the manuscript: None, since the point is beyond the scope of the manuscript.

That having been said, the paper contains a number of interesting and valuable design points that certainly deserve further discussion. To guide that discussion better I recommend to account for the above suggestions and questions.

**Answer: We like to thank the reviewer for the positive comment and for pointing out unclear points in our manuscript. Of course we tried to do our best to answer all questions and to improve the text. We trust that our answers to the reviewer's questions are satisfactory.**

---

## Author Response (AR1)

Dear Editor,

The revised version of our manuscript was prepared taking into account the two anonymous reviewers' comments as detailed in our responses. The changes to the text are marked in red, we Fig. 7 was split into two figures (7a and 7b). We added 7 new references and corrected a number of minor errors not mentioned by the reviewers.

With very best regards
Ulrich Platt